# DisCO: Reinforcing Large Reasoning Models with Discriminative Constrained Optimization

Gang Li[1]    Ming Lin    Tomer Galanti[1]    Zhengzhong Tu[1]    Tianbao Yang[1]

[1] Texas A&M University

{gang-li, galanti, tzz, tianbao-yang}@tamu.edu

linming04@gmail.com

## Abstract

The recent success and openness of DeepSeek-R1 have brought widespread attention to Group Relative Policy Optimization (GRPO) as a reinforcement learning method for large reasoning models (LRMs). In this work, we analyze the GRPO objective under a binary reward setting and reveal an inherent limitation of question-level difficulty bias arising from its group relative advantage function. We also identify a connection between GRPO and traditional discriminative methods in supervised learning. Motivated by these insights, we introduce a new **Discriminative Constrained Optimization (DisCO)** framework for reinforcing LRMs, grounded in the principle of discriminative learning: increasing the scores of positive answers while decreasing those of negative ones. The main differences between DisCO and GRPO and its recent variants are: (1) it replaces the group relative objective with a discriminative objective defined by a scoring function; (2) it abandons clipping-based surrogates in favor of non-clipping RL surrogate objectives used as scoring functions; (3) it employs a simple yet effective constrained optimization approach to enforce the KL divergence constraint. As a result, DisCO offers notable advantages over GRPO and its variants: (i) it completely eliminates difficulty bias by adopting discriminative objectives; (ii) it addresses the entropy instability in GRPO and its variants through the use of non-clipping scoring functions and a constrained optimization approach, yielding long and stable training dynamics; (iii) it allows the incorporation of advanced discriminative learning techniques to address data imbalance, where a significant number of questions have more negative than positive generated answers during training. Our experiments on enhancing the mathematical reasoning capabilities of SFT-finetuned models show that DisCO significantly outperforms GRPO and its improved variants such as DAPO, achieving average gains of 7% over GRPO and 6% over DAPO across six benchmark tasks for a 1.5B model.[1]

## 1 Introduction

The recent success and openness of DeepSeek-R1 have sparked a surge of interest in large reasoning models (LRMs), particularly in the context of fine-tuning via reinforcement learning (RL) [23]. The core approach involves iteratively generating synthetic data using the reasoning model and applying a rule-based reward mechanism to label the outputs. These rewards are then used to update the policy, *i.e.*, the reasoning model itself. Notably, this framework, featuring a novel policy optimization method called Group Relative Policy Optimization (GRPO), has enabled DeepSeek-R1 to achieve performance comparable to advanced proprietary LRMs at that time such as OpenAI-o1 on many reasoning benchmarks. As a result, GRPO has rapidly become a focal point for advancing LRM capabilities, particularly in domains like mathematics and scientific reasoning.

---

[1]The code is available at: `https://github.com/Optimization-AI/DisCO`

39th Conference on Neural Information Processing Systems (NeurIPS 2025).

Several efforts have sought to replicate the performance of DeepSeek-R1 or to further enhance reasoning models using GRPO [67, 31, 42, 27, 69, 74, 5], while few others have tried to identify its inherent limitations with potential remedies [40, 79, 39]. Useful tricks have been introduced to improve GRPO [40, 41, 79, 27, 13, 82]. For instance, DAPO [79] employs two distinct clipping hyperparameters to mitigate *entropy collapse*, encouraging exploration. Dr. GRPO [40] removes the variance normalization in advantage function, aiming to mitigate the issue of *difficulty bias*. However, these approaches remain heuristic and ad-hoc, lacking a principled foundation and falling short of fully addressing GRPO's inherent limitations. Our analysis identifies that Dr. GRPO continues to suffer from the difficulty bias issue, while our experiments show that DAPO may induce excessive entropy growth, producing highly random outputs. This motivates us to explore a central question:

> *How can we design more effective optimization methods for reinforcing large reasoning models in a principled manner without inheriting the limitations of GRPO?*

This paper addresses the above question through a complete redesign of the objective function, grounded in the principles of discriminative learning. Specifically, we first analyze the objective function of GRPO and its variants under **a binary reward setting**, leading to two key insights: **(1)** the root cause of GRPO's difficulty bias lies in its group relative advantage function, which induces disproportionately small weights to questions that are either too easy or too hard; and **(2)** there exists a conceptual connection to traditional discriminative approaches in AUC maximization, which aim to increase the scores of positive outputs while decreasing that of negative outputs.

Building upon these insights, we propose a principled optimization framework for reinforcing large reasoning models based on discriminative learning. Specifically, we optimize a discriminative objective using a proper scoring function over input-output pairs, which increases the score of positive outputs and decreases that of negative ones. The flexibility of our framework allows us to leverage simple non-clipping RL surrogate objectives as scoring functions without suffering from entropy instability, and to incorporate advanced discriminative techniques to address data imbalance in generated rollouts. To ensure training stability, we adopt a simple yet effective constrained optimization method to enforce a trust region constraint bounding the KL divergence between the updated model and the old model. Our experiments for mathematical reasoning show that DisCO significantly outperforms all baselines for fine-tuning DeepSeek-R1-Distill-Qwen and -Llama models with a maximum 8k response length for both training and inference, and also achieves a better performance than GRPO that uses a maximum 24k length for training and 32k length for inference.

Our main contributions are summarized as follows:

- We present an **analysis of GRPO's objective function**, identifying the root cause of difficulty bias and revealing its conceptual connection to classic discriminative methods for AUC maximization.
- We introduce a **principled discriminative constrained optimization framework** for reinforcing large reasoning models, which avoids both difficulty bias and training instability. This framework gives rise to a family of methods we refer to as **DisCO**.
- We demonstrate **significant improvements** of our DisCO method over GRPO and four other baselines, including DAPO, through experiments for fine-tuning LRMs on mathematical reasoning tasks, with evaluations across six benchmarks.

## 2 Related Work

**Large Reasoning Models (LRMs).** Recent advances of LRMs, such as OpenAI o1 [51], DeepSeek-R1 [23] and Kimi K1.5 [63], have demonstrated strong reasoning capability in solving complex tasks. Departing from earlier approaches in LLMs, such as Chain-of-thought (CoT) prompting [66, 48, 81], Tree-of-Thought [78], Monte Carlo Tree Search [18, 64, 71], a major breakthrough was achieved by scaling RL training using verifiable rewards to incentivize LLMs to learn through self-exploration [23, 63]. Inspired by DeepSeek-R1's core algorithm GRPO [59], the research community has actively pursued improved techniques for large-scale RL training, focusing primarily on three directions: algorithm design [79, 40, 13, 39, 63, 61], reward curation [82, 67, 79], and sampling strategies [79, 27, 83, 30]. Our work falls under the category of algorithm design.

Among these, Dr. GRPO [40] identifies response-level length bias and question-level difficulty bias in GRPO algorithm, advocating the removal of length and advantage normalization to improve token efficiency. DAPO [79] highlights several limitations of GRPO, such as entropy collapse, training instability, and biased loss, and addresses them through techniques like decoupled clipping,

dynamic sampling, and a token-level policy loss. GPG [13] introduces a simplified REINFORCE-based objective that eliminates the need for both the critic and reference models, thereby enhancing scalability for RL training. TRPA [61] simply uses the Direct Preference Optimization (DPO) objective and a KL divergence regularization for fine-tuning LRMs. It can be recovered from our basic approach, which uses a logistic function as the surrogate loss, the log of likelihood ratio with respect to a frozen reference model as the scoring function, and the KL divergence as a regularization rather than a constraint. However, it does not address the imbalanced rollouts. The uniqueness and significance of our contributions lie in the analysis of GRPO objective and its variants that reveal key limitations, and the integration of advanced discriminative learning approaches for handling imbalanced rollouts and efficient constrained optimization technique for ensuring training stability.

**Reinforcement Learning (RL).** RL is a learning paradigm centered on control and decision-making, in which an agent optimizes a target objective through trial-and-error interactions with its environment [8]. RL approaches are typically categorized into model-based [60, 53, 49, 17] and model-free methods [68, 62, 46, 56, 57, 38, 20]. Among model-free methods, the evolution from Vanilla Policy Gradient [68, 62] to TRPO [56] and PPO [57] has influenced the development of GRPO. In the context of fine-tuning LLMs, another line of work is RL from human feedback (RLHF). An early example of connecting RL with LLMs dates back to OpenAI's work on integrating human preferences to improve text generation tasks, such as summarization using the PPO algorithm [85]. This approach was later extended to fine-tune LLMs for instruction following and/or alignment on helpfulness and harmlessness [52, 3, 22, 75]. Due to the high data requirements and training costs of standard RLHF, off-policy methods like DPO [54] and its variants [2, 16, 72, 44, 24], have been proposed to reduce reliance on explicit reward models. Another line of on-policy algorithms for RLHF, such as RLOO [1], ReMax [36], and REINFORCE++ [29], has been introduced to reduce the computational burden by removing the critic network in PPO. While some works [43, 32, 29, 10, 70] attempt to adapt RLHF techniques for reasoning tasks, they have not yielded significant improvements.

**Discriminative Learning.** Parallel to RL, discriminative learning is another classical learning paradigm, that has been studied extensively for many traditional tasks, including multi-class classification [14, 15, 6], AUC maximization [76, 80], and learning to rank [9, 19, 7]. These methods are grounded in the common principle of increasing prediction scores for positive (relevant) labels (data) while decreasing scores for negative (irrelevant) ones. Nevertheless, discriminative learning remains under-explored in the cotext of LLM training. Recently, Guo et al. [25] proposed discriminative probabilistic approaches for supervised fine-tuning of LLMs. However, unlike our approach, they did not employ an RL framework with verifiable rewards to fine-tune LRMs.

## 3 Preliminaries

We consider fine-tuning a generative reasoning model $\pi_\theta$ parameterized by $\theta$. The old model in one step of learning is denoted by $\pi_{\text{old}}$. It is used to generate answers for a set of input questions. Given a question $q$ (with prompt included), the generated output $o$ follows the distribution $\pi_{\text{old}}(\cdot|q)$, which includes reasoning traces and the final answer. Specifically, output $o$ is generated token by token, i.e., $o_t \sim \pi_{\text{old}}(\cdot|q, o_{<t})$, for $t = 1, \cdots, |o|$. We consider a rule-based reward mechanism that returns a binary value for a given question $q$ and its corresponding answer in the output $o$, which uses either exact match against extracted answer or a formal verification tool [23, 33, 55]. Let $r(o|q) \in \{1, 0\}$ denote the reward assigned to an output $o$ with respect to the input $q$. Let $p(q) = \mathbb{E}_{o\sim\pi_{\text{old}}(\cdot|q)}[r(o|q)] \in [0, 1]$, which quantifies the difficulty of the question $q$ under the model $\pi_{\text{old}}$. We denote by $\pi_{\text{old}}^+(\cdot|q)$ the conditional distribution of outputs when the reward is one (i.e., positive answers) and by $\pi_{\text{old}}^-(\cdot|q)$ the conditional distribution of outputs when the reward is zero (i.e., negative answers). By the law of total expectation, for any function $g(o, q)$ we have

$$\mathbb{E}_{o\sim\pi_{\text{old}}(\cdot|q)}[g(o, q)] = p(q)\mathbb{E}_{o\sim\pi_{\text{old}}^+(\cdot|q)}[g(o, q)] + (1 - p(q))\mathbb{E}_{o\sim\pi_{\text{old}}^-(\cdot|q)}[g(o, q)]. \quad (1)$$

**Group Relative Policy Optimization (GRPO).** The key idea of GRPO is to generate multiple outputs for an input $q$ and define a group relative advantage function. For analysis, we consider the expectation formulation instead of empirical average of the GRPO objective for maximization:

$$\mathcal{J}_{\text{GRPO}}(\theta) = \mathbb{E}_q \mathbb{E}_{o\sim\pi_{\text{old}}(\cdot|q)} \left[ \frac{1}{|o|} \sum_{t=1}^{|o|} f\left( \frac{\pi_\theta(o_t|q, o_{<t})}{\pi_{\text{old}}(o_t|q, o_{<t})}, A(o|q) \right) \right] - \beta \mathbb{D}_{\text{KL}}(\pi_\theta || \pi_{\text{ref}}), \quad (2)$$

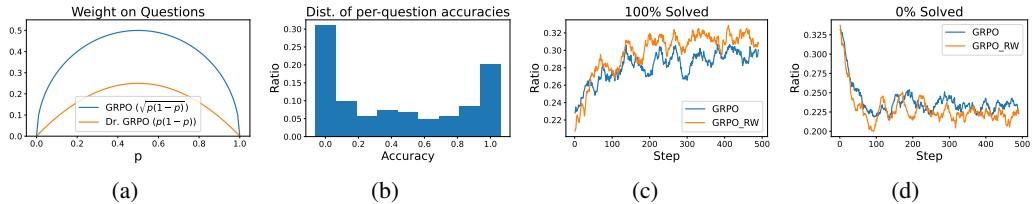

Figure 1: (a) Weight on questions based on correctness probability $p$; (b) Histogram of per-question accuracy evaluated in the GRPO learning; (c) Comparison of the ratio of questions with 100% correctness probability; (d) Comparison of the ratio of questions with 0% correctness probability.

where $f(x,y) = \min(xy, \text{clip}(x, 1-\epsilon, 1+\epsilon)y)$, $A(o|q) = \frac{(r(o|q) - \mathbb{E}_{o' \sim \pi_{\text{old}}(\cdot|q)} r(o'|q))}{\sqrt{\text{Var}_{o' \sim \pi_{\text{old}}(\cdot|q)} r(o'|q)}}$ is the advantage function that quantifies how much better the reward of $o$ is compared to average reward, $\pi_{\text{ref}}$ is a frozen reference model.

Recently, several variants of GRPO have been introduced [79, 13, 39, 82, 40]. Many of them retain the advantage function $A(o|q)$ while modifying other components such as hyper-parameter $\epsilon$, the normalization factor and the likelihood ratio. Several works employ an unnormalized advantage function $\hat{A}(o|q) = r(o|q) - \mathbb{E}_{o' \sim \pi_{\text{old}}(\cdot|q)} r(o'|q)$ [40, 13].

## 4 Analysis of GRPO and its variants

In the following analysis we assume $p(q) \in (0,1)$; otherwise we can remove them from consideration as done in practice [23, 42, 59].

**Proposition 1** *Let us consider the objective of GRPO and its variants with the following form:*

$$\mathcal{J}_0(\theta) = \mathbb{E}_q \mathbb{E}_{o \sim \pi_{old}(\cdot|q)} \left[ \frac{1}{|o|} \sum_{t=1}^{|o|} f\left( \frac{\pi_\theta(o_t|q, o_{<t})}{\pi_{old}(o_t|q, o_{<t})}, A(o|q) \right) \right]. \tag{3}$$

*Assume that $f(x,y)$ is non-decreasing function of $x$ such that $f(x,y) = \mathbb{I}(y>0)yf^+(x,1) - \mathbb{I}(y \leq 0)|y|f^-(x,1)$, where both $f^+, f^-$ are non-decreasing functions of $x$, then we have*

$$\mathcal{J}_0(\theta) = \mathbb{E}_q \sqrt{p(q)(1-p(q))} \mathbb{E}_{o \sim \pi_{old}^+(\cdot|q), o' \sim \pi_{old}^-(\cdot|q)} [s_\theta^+(o,q) - s_\theta^-(o',q)], \tag{4}$$

*where $s_\theta^+(o,q) = \frac{1}{|o|} \sum_{t=1}^{|o|} f^+\left( \frac{\pi_\theta(o_t|q, o_{<t})}{\pi_{old}(o_t|q, o_{<t})}, 1 \right)$ and $s_\theta^-(o,q) = \frac{1}{|o|} \sum_{t=1}^{|o|} f^-\left( \frac{\pi_\theta(o_t|q, o_{<t})}{\pi_{old}(o_t|q, o_{<t})}, 1 \right)$. In particular, for GRPO we have*

$$f^+(x,1) = \min(x, 1+\epsilon), \quad f^-(x,1) = \max(x, 1-\epsilon). \tag{5}$$

**Remark:** The assumption of $f(x,y)$ indeed holds for GRPO and its variants. We will present the analysis for several variants of GRPO in Appendix B.3.

The proof of the above proposition is included in Appendix B.1 and is inspired by [47] with differences that lead to two **new insights** from Proposition 1 regarding the two components of $\mathcal{J}_0$. First, let us consider the component $\mathbb{E}_{o \sim \pi_{\text{old}}^+(\cdot|q), o' \sim \pi_{\text{old}}^-(\cdot|q)}[s_\theta^+(o,q) - s_\theta^-(o',q)]$. Since both $f^+$ and $f^-$ are non-decreasing functions of the first argument, then both $s_\theta^+(o,q)$ and $s_\theta^-(o,q)$ are non-decreasing functions of $\pi_\theta(o_t|q, o_{<t})$. Hence, maximizing $\mathcal{J}_0$ would increase the likelihood of tokens in the positive answers and decrease the likelihood of tokens in the negative answers. This makes sense as we would like the new model to have a high likelihood of generating a positive (correct) answer and a low likelihood of generating a negative (incorrect) answer. This mechanism is closely related to traditional discriminative methods of supervised learning in the context of AUC maximization [77], which aims to maximize the scores of positive samples $o \sim \pi_{\text{old}}^+(\cdot|q)$ while minimizing scores of negative samples $o' \sim \pi_{\text{old}}^-(\cdot|q)$, where the $q$ acts like the classification task in the AUC maximization. Hence, in the context of discriminative learning, we refer to $s^+(o,q)$ and $s^-(o,q)$ as scoring functions. Therefore, $\mathbb{E}_{o \sim \pi_{\text{old}}^+(\cdot|q), o' \sim \pi_{\text{old}}^-(\cdot|q)}[s^+(o,q) - s^-(o',q)]$ is a discriminative objective.

Second, let us consider the component $\omega(q) = \sqrt{p(q)(1-p(q))}$, which acts like a weight scaling the discriminative objective for each individual input question. It is this component that leads to difficulty bias. As shown in Figure 1(a), questions with very high $p(q)$ values (close to 1) or very

low $p(q)$ values (close to 0) receive small weights for their discriminative objectives, causing the optimization to focus primarily on questions of intermediate difficulty while paying little attention to hard questions ($p(q) \approx 0$) and easy questions ($p(q) \approx 1$). This mechanism may significantly hinder the learning efficiency. Intuitively, if the generated answers have only one correct solution out of 10 trials, i.e. $p(q) = 0.1$, we should grasp this chance to enhance the model instead of overlooking it. On the other hand, even when we encounter an easy question with a probability of $p(q) = 0.9$, we should keep improving the model rather than being satisfied because it still makes mistakes with respect to this question. Our hypothesis is that removing this weight could accelerate the training. To validate this hypothesis, we conducted a series of empirical experiments for fine-tuning a 1.5B model as described in Section 6. We start by examining whether a substantial number of questions have correctness probabilities ($p(q)$) near 0 or 1. As shown in Figure 1(b), during GRPO training, the correctness probabilities across individual questions appear broadly distributed, with many near 0 or 1. Then, we compare the original GRPO with a variant that removes weight $\sqrt{p(q)(1-p(q))}$:

$$\mathcal{J}_{\text{GRPO\_RW}} = \mathbb{E}_q \mathbb{E}_{o \sim \pi^+_{\text{old}}, o' \sim \pi^-_{\text{old}}} [s^+(o, q) - s^-(o', q)] - \beta \mathbb{D}_{\text{KL}}(\pi_\theta || \pi_{ref}). \tag{6}$$

The results are shown in Figure 1(c) and 1(d). We can observe that the variant without the weighting mechanism quickly achieves a higher ratio of 100% correctness and a lower ratio of 0% correctness, confirming the detrimental impact of the inappropriate weighting.

We note that the difficulty bias has been pointed out in a recent work Dr. GRPO [40]. To mitigate this issue, Dr. GRPO uses the un-normalized advantage function $\hat{A}(o|q)$. However, with a similar analysis as above (cf. Appendix B.3), we can derive that Dr. GRPO still has a question-level weight $\omega(q) = p(q)(1 - p(q))$ before the discriminative objective. As shown in Figure 1(a), this weight mitigates but does not eliminate the imbalanced weight across questions.

# 5 A Discriminative Constrained Optimization Framework

While the last section has suggested a tangible remedy to address the difficulty bias of GRPO and its variants by removing the weight before the discriminative objective, there are other issues of the scoring function of GRPO and its variants. Next, we propose a general discriminative learning framework for reinforcing LRMs and incorporate advanced techniques to facilitate the learning.

## 5.1 A basic approach

Motivated by the connection with AUC maximization, we redesign the objective directly from the principle of discriminative learning. For a given question $q$, let $s_\theta(o, q)$ denote a scoring function that measures how likely the model $\pi_\theta$ "predicts" the output $o$ for a given input $q$ [2]. Then the AUC score for the "task" $q$ is equivalent to $\mathbb{E}_{o \sim \pi^+_{\text{old}}, o' \sim \pi^-_{\text{old}}} [\mathbb{I}(s_\theta(o, q) > s_\theta(o', q))]$. Using a continuous surrogate function $\ell$, we form the following objective (in expectation form) for maximization:

$$\mathcal{J}_1(\theta) = \mathbb{E}_q \mathbb{E}_{o \sim \pi^+_{\text{old}}(\cdot|q), o' \sim \pi^-_{\text{old}}(\cdot|q)} \ell(s_\theta(o, q) - s_\theta(o', q)). \tag{7}$$

Different surrogate functions $\ell(\cdot)$ can be used. For comparison, with GRPO, we simply use the identity function $\ell(s) = s$. One difference from the discriminative objective (6) is that we use a single scoring function $s_\theta(o, q)$ for both positive outputs $o$ and negative outputs $o'$. It is notable that the different scoring functions for positive and negative outputs in (6) actually arise from the clipping operations of GRPO objective. Recent works have found that the clipping could lead to entropy collapse [79]. In addition, the clipping could cause the vanishing gradient, which may also slow down the learning process. To avoid these issues, we consider non-clipping scoring functions.

**Scoring functions.** We consider two choices of scoring functions, i.e., log-likelihood and likelihood ratio. The log-likelihood (log-L) scoring function is defined by $s_\theta(o, q) = \frac{1}{|o|} \sum_{t=1}^{|o|} \log \pi_\theta(o_t | q, o_{<t})$. The likelihood ratio (L-ratio) scoring function is computed by $s_\theta(o, q) = \frac{1}{|o|} \sum_{t=1}^{|o|} \frac{\pi_\theta(o_t | q, o_{<t})}{\pi_{\text{old}}(o_t | q, o_{<t})}$. In Appendix B.2, we discuss the connection between the two scoring functions and the surrogate objectives of vanilla policy gradient methods [68] and TRPO [56], respectively.

**Stabilize the training with Constrained Optimization.** Training instability is a long-standing issue in RL [56, 57]. Different methods have been introduced to ensure stability. Recent RL-based methods for learning reasoning models either follow the clipping operation of PPO [57] or use the KL divergence regularization $\mathbb{D}_{\text{KL}}(\pi_\theta || \pi_{\text{ref}})$ or $\mathbb{D}_{\text{KL}}(\pi_{\text{old}} || \pi_\theta)$ [59, 61, 52]. However, the clipping

---

[2] in the context of generative models, "predicts" is like "generates".

operation could lead to the entropy collapse [79], which we try to avoid by using the non-clipped scoring function. The KL divergence regularization $\mathbb{D}_{\mathrm{KL}}(\pi_\theta||\pi_{\mathrm{ref}})$ while being used in traditional RL is not effective for preventing entropy collapse [27, 79]. Regarding the regularization with $\mathbb{D}_{\mathrm{KL}}(\pi_{\mathrm{old}}||\pi_\theta)$, earlier studies [56, 57] has found that it would be difficult to choose a single value of the regularization parameter that performs well across different problems or even within a single problem where the the characteristics change over the course of learning. To tackle this issue, we revisit the idea of trust region constraint of TRPO [56], i.e., restricting the updated model $\theta$ in the trust region $\mathbb{D}_{\mathrm{KL}}(\pi_{\mathrm{old}}||\pi_\theta) \leq \delta$. As a result, we solve the following **discriminative constrained optimization** problem:

$$\max_\theta \mathcal{J}_1(\theta) := \mathbb{E}_q \mathbb{E}_{o \sim \pi_{\mathrm{old}}^+(\cdot|q), o' \sim \pi_{\mathrm{old}}^-(\cdot|q)} \ell(s_\theta(o, q) - s_\theta(o', q))$$
$$s.t. \quad \mathbb{D}_{\mathrm{KL}}(\pi_{\mathrm{old}}||\pi_\theta) \leq \delta. \tag{8}$$

For sake of efficiency, we use a different optimization approach from TRPO to solve the above constrained optimization. Inspired by the recent advances of non-convex inequality constrained optimization algorithm [35], we adopt a squared-hinge penalty function for the constraint and solve the following problem with an appropriate penalty parameter $\beta$:

$$\max_\theta \mathbb{E}_q \mathbb{E}_{o \sim \pi_{\mathrm{old}}^+(\cdot|q), o' \sim \pi_{\mathrm{old}}^-(\cdot|q)} \ell(s_\theta(o, q) - s_\theta(o', q)) - \beta[\mathbb{D}_{\mathrm{KL}}(\pi_{\mathrm{old}}||\pi_\theta) - \delta]_+^2, \tag{9}$$

where $[\cdot]_+ = \max\{\cdot, 0\}$. It has been shown that under an appropriate assumption regarding the constraint function and $\beta$, solving the above squared-hinge penalized objective (9) can return a KKT solution of the original constrained problem (8). We refer the readers to [35] for more in-depth analysis of this approach.

Finally, we would like to emphasize the difference between using the squared-hinge penalty function and the regular KL divergence regularization $\beta\mathbb{D}_{\mathrm{KL}}(\pi_{\mathrm{old}}||\pi_\theta)$. The squared-hinge penalty function has a dynamic weighting impact for the gradient, $\nabla\beta[\mathbb{D}_{\mathrm{KL}}(\pi_{\mathrm{old}}||\pi_\theta) - \delta]_+^2 = 2\beta[\mathbb{D}_{\mathrm{KL}}(\pi_{\mathrm{old}}||\pi_\theta) - \delta]_+ \nabla\mathbb{D}_{\mathrm{KL}}(\pi_{\mathrm{old}}||\pi_\theta)$, such that if the constraint is satisfied then the weight $2\beta[\mathbb{D}_{\mathrm{KL}}(\pi_{\mathrm{old}}||\pi_\theta) - \delta]_+$ before the gradient of the regularization term $\mathbb{D}_{\mathrm{KL}}(\pi_{\mathrm{old}}||\pi_\theta)$ becomes zero. This means the KL divergence regularization is only effective when the constraint is violated. In contrast, the regular KL divergence regularization $\beta\mathbb{D}_{\mathrm{KL}}(\pi_{\mathrm{old}}||\pi_\theta)$ always contributes a gradient $\beta\nabla\mathbb{D}_{\mathrm{KL}}(\pi_{\mathrm{old}}||\pi_\theta)$ no matter whether the constraint is satisfied or not, which could harm the learning.

## 5.2 An improved approach for tackling imbalanced rollouts

One advantage of designing the objective based on the principle of discriminative learning is the ability to leverage a wide range of advanced techniques from the literature to improve training. A key challenge in RL fine-tuning for reasoning models is the sparse rewards, which lead to imbalance in generated rollouts. Specifically, for some questions where $p(q) \ll 1$, the number of negative outputs can significantly exceed the number of positive ones. This reflects a classic data imbalance issue, which has been extensively studied in the discriminative learning community [84, 58, 50]. To address this issue, we consider distributionally robust optimization (DRO) [84, 50].

Let us first discuss why the basic approach could be ineffective for combating the imbalanced rollouts. The objective function $\mathcal{J}_1$ is motivated by maximizing AUC for each question $q$, i.e., $\mathbb{E}_{o \sim \pi_{\mathrm{old}}^+, o' \sim \pi_{\mathrm{old}}^-}[\mathbb{I}(s_\theta(o, q) > s_\theta(o', q))]$. However, when there is much more negative data than positive data, AUC is not a good measure. For example, let us consider a scenario where there are 1 positive $o_+$ and 100 negatives $\{o_-^1, \ldots, o_-^{100}\}$. If the scores of these data are $s(o_-^1, q) = 0.9, s(o_+, q) = 0.5, s(o_-^2, q) = s(o_-^3, q) \ldots = s(o_-^{100}, q) = 0.001$, then the AUC score is $\frac{99}{100} = 0.99$. The AUC score is high but is not informative as the model still generates the negative data $o_-^1$ more likely than the positive data $o_+$. In the literature, this issue has been addressed by maximizing a partial AUC score, which considers the pairwise order between all positives and the top ranked negatives. We utilize a surrogate function of partial AUC score formulated from the perspective of DRO [84].

Consider a question $q$ and a positive data $o$. We denote by $\mathcal{Q}$ the set of probability measures $Q$ on negative data given $q$ (absolutely continuous with respect to $\pi_{\mathrm{old}}^-(\cdot|q)$). Denote by $\mathbb{D}_{\mathrm{KL}}(Q, \pi_{\mathrm{old}}^-(\cdot|q))$ the KL divergence between a distribution $Q$ and the negative data distribution $\pi_{\mathrm{old}}^-(\cdot|q)$. A DRO

Table 1: Comparison of different methods for reinforcing large reasoning models. "L-ratio" means likelihood ratio, "log-L" means log-likelihood, "proper" means any proper scoring function.

| Method | Difficulty Bias | Clipping | KL Divergence | Score Function | Tackles Imbalanced Rollouts |
|---|---|---|---|---|---|
| GRPO [23] | Yes | Yes | regularization, $\pi_{\text{ref}}$ | clipped L-ratio | No |
| Dr. GRPO [40] | Yes | Yes | No | clipped L-ratio | No |
| DAPO [79] | Yes | Yes | No | clipped L-ratio | No |
| GPG [13] | Yes | No | No | log-L | No |
| TRPA [61] | No | No | regularization, $\pi_{\text{old}}$ | log L-ratio | No |
| DisCO | No | No | constraint, $\pi_{\text{old}}$ | proper | Yes |

formulation for partial AUC maximization is given by [84][Theorem 2]:

$$\inf_{Q \in \mathcal{Q}} \tau \mathbb{D}_{\text{KL}}(Q, \pi_{\text{old}}^-(\cdot|q)) + \mathbb{E}_{o' \sim Q}[s_\theta(o, q) - s_\theta(o', q)]:$$

$$= -\tau \log \left( \mathbb{E}_{o' \sim \pi_{\text{old}}^-(\cdot|q)} \exp \left( \frac{s_\theta(o', q) - s_\theta(o, q)}{\tau} \right) \right).$$

As a result, we construct the following DRO-based objective for maximization:

$$\mathcal{J}_2(\theta) = -\mathbb{E}_q \mathbb{E}_{o \sim \pi_{\text{old}}^+(\cdot|q)} \tau \log \left( \mathbb{E}_{o' \sim \pi_{\text{old}}^-(\cdot|q)} \exp \left( \frac{s_\theta(o', q) - s_\theta(o, q)}{\tau} \right) \right). \tag{10}$$

It is easy to show that $\mathcal{J}_2(\theta) \leq \mathcal{J}_1(\theta)$ by Jensen's inequality for the convex function $-\log$. Hence, maximizing $\mathcal{J}_2(\theta)$ will automatically increasing $\mathcal{J}_1(\theta)$. However, the reverse is not true. This also explains why maximizing $\mathcal{J}_2(\theta)$ could be more effective than maximizing $\mathcal{J}_1(\theta)$. The above risk function is also known as optimized certainty equivalents (OCE) in mathematical finance [4]. We would like to point out that although OCE or DRO has been considered for RL in existing works [65, 73], they differ from our work for addressing different issues. Wang et al. [65] apply the OCE to compute a robust reward, replacing the standard expected reward used in traditional RL settings. Xu et al. [73] adopt DRO for direct preference optimization of LLMs, aiming to mitigate the noise in human preference data by addressing the distributional shift between the empirical distribution of $(q, o, o')$ and its true underlying distribution.

Finally, we solve the following discriminative constrained optimization problem by using the same squared-hinge penalty method:

$$\max_\theta \mathcal{J}_2(\theta) := -\mathbb{E}_q \mathbb{E}_{o \sim \pi_{\text{old}}^+(\cdot|q)} \tau \log \left( \mathbb{E}_{o' \sim \pi_{\text{old}}^-(\cdot|q)} \exp \left( \frac{s_\theta(o', q) - s_\theta(o, q)}{\tau} \right) \right),$$

$$s.t. \quad \mathbb{D}_{\text{KL}}(\pi_{\text{old}}||\pi_\theta) \leq \delta. \tag{11}$$

To differentiate the approach for solving (8) and (11), we refer to the former as **DisCO-b** and the latter as **DisCO**. In practice, all expectations will be replaced by empirical averages and the KL divergence is also estimated at each iteration by using sampled data following [52]. We present a full algorithm in Algorithm 1 in Appendix. Finally, we give a comparison between DisCO and existing RL fine-tuning methods for reinforcing LRMs from different aspects in Table 1.

## 6 Experiments

In this section, we empirically evaluate the effectiveness of the proposed DisCO by comparing with GRPO and other variants for reinforcing SFT-finetuned models.

**Task Setting.** We validate our method on mathematical reasoning tasks. Specifically, we use the DeepScaleR-Preview-Dataset [42] for training, which includes AIME problems from 1984 to 2023, AMC problems before 2023, and questions from the Omni-MATH [21] and Still [45] datasets, totaling approximately 40.3k unique problem-answer pairs. We evaluate models on six benchmark datasets: AIME 2024, AIME 2025, MATH 500 [28, 37], AMC 2023, Minerva [34], and Olympiad Bench (O-Bench) [26]. Following [23, 42], we adopt the pass@1 metric [11] averaged over $k = 16$ responses for each question to ensure the reliability of model performances. The metric for each question is calculated as $\frac{1}{k} \sum_{i=1}^k \mathbb{I}(o_i \text{ is correct})$, where $o_i$ denotes the $i$-th generated response. For both the training and evaluation of our method and the baselines (unless otherwise specified), the maximum response length is limited to 8k tokens. To verify the generalizability of our method to other datasets, we also conducted experiments on DAPO-Math-17k [79] dataset, which is included in the Appendix A.3.

Table 2: Comparison with baseline models and baseline methods for fine-tuning 1.5B models. OpenAI-o1-preview is included as a reference. MRL denotes Max Response Length utilized in training/testing. The shaded models are trained by other works and the shaded numbers are reported in their original works or in [42]. All other results are either evaluated on existing models or on the models trained by us using different approaches. Methods in the bottom area are all for fine-tuning DeepSeek-R1-Distill-Qwen-1.5B model on the same DeepScaleR dataset. DS is short for DeepSeek-R1, DSR is short for DeepScaleR.

| Model/Method | MRL(Train/Test) | AIME 2024 | AIME 2025 | MATH 500 | AMC 2023 | Minerva | O-Bench | Avg. |
|---|---|---|---|---|---|---|---|---|
| OpenAI-o1-Preview | - | 0.4 | - | 0.814 | - | - | - | - |
| DS-Distill-Qwen-1.5B | 32k+ / 32k | 0.288 | 0.263 | 0.828 | 0.629 | 0.265 | 0.433 | 0.451 |
| DS-Distill-Qwen-1.5B | 32k+ / 8k | 0.181 | 0.215 | 0.758 | 0.515 | 0.237 | 0.353 | 0.376 |
| STILL-3-1.5B-preview | 29k / 32k | 0.325 | 0.248 | 0.844 | 0.667 | 0.290 | 0.454 | 0.471 |
| DSR-1.5B-Preview | 24k / 32k | **0.431** | 0.304 | **0.878** | 0.736 | 0.302 | 0.500 | 0.525 |
| DSR-1.5B-Preview | 24k / 8k | 0.358 | 0.258 | 0.860 | 0.679 | 0.297 | 0.473 | 0.488 |
| GRPO | 8k / 8k | 0.277 | 0.242 | 0.838 | 0.647 | 0.276 | 0.462 | 0.457 |
| GRPO-ER | 8k / 8k | 0.298 | 0.242 | 0.839 | 0.649 | 0.279 | 0.452 | 0.460 |
| Dr. GRPO | 8k / 8k | 0.252 | 0.238 | 0.831 | 0.631 | 0.268 | 0.440 | 0.443 |
| DAPO | 8k / 8k | 0.310 | 0.252 | 0.848 | 0.675 | 0.296 | 0.456 | 0.473 |
| TRPA | 8k / 8k | 0.354 | 0.235 | 0.835 | 0.653 | 0.283 | 0.458 | 0.470 |
| DisCO (L-ratio) | 8k / 8k | 0.381 | 0.306 | **0.878** | 0.746 | 0.319 | **0.512** | 0.524 |
| DisCO (log-L) | 8k / 8k | 0.404 | **0.317** | 0.876 | **0.758** | **0.333** | 0.509 | **0.533** |

Table 3: Comparison with baseline models and baseline methods for fine-tuning 7B models. Methods in the bottom area are all for fine-tuning DeepSeek-R1-Distill-Qwen-7B model on the same DeepScaleR dataset.

| Model/Method | MRL(Train/Test) | AIME 2024 | AIME 2025 | MATH 500 | AMC 2023 | Minerva | O-Bench | Avg. |
|---|---|---|---|---|---|---|---|---|
| DS-Distill-Qwen-7B | 32k+ / 32k | 0.560 | 0.396 | 0.923 | 0.825 | 0.380 | 0.568 | 0.609 |
| DS-Distill-Qwen-7B | 32k+ / 8k | 0.402 | 0.292 | 0.873 | 0.688 | 0.355 | 0.471 | 0.513 |
| GRPO-LEAD-7B | 8k / 8k | 0.470 | 0.345 | 0.893 | 0.748 | 0.372 | 0.500 | 0.555 |
| TRPA | 8k / 8k | 0.570 | - | 0.870 | 0.780 | 0.360 | 0.550 | - |
| GRPO | 8k / 8k | 0.498 | 0.394 | 0.916 | 0.807 | 0.381 | 0.555 | 0.592 |
| GRPO-ER | 8k / 8k | 0.515 | 0.381 | 0.916 | 0.825 | 0.376 | 0.544 | 0.593 |
| Dr. GRPO | 8k / 8k | 0.488 | 0.346 | 0.910 | 0.792 | 0.368 | 0.546 | 0.575 |
| DAPO | 8k / 8k | 0.454 | 0.335 | 0.907 | 0.799 | 0.388 | 0.535 | 0.570 |
| TRPA | 8k / 8k | 0.510 | 0.367 | 0.898 | 0.779 | 0.379 | 0.534 | 0.578 |
| DisCO (L-ratio) | 8k / 8k | **0.583** | **0.421** | 0.923 | 0.852 | 0.399 | 0.585 | **0.627** |
| DisCO (log-L) | 8k / 8k | 0.558 | 0.410 | **0.927** | **0.854** | **0.410** | **0.592** | 0.625 |

Table 4: Comparison with baseline models and baseline methods for fine-tuning 8B models. Methods in the bottom area are all for fine-tuning DeepSeek-R1-Distill-Llama-8B model on the same DeepScaleR dataset.

| Model/Method | MRL(Train/Test) | AIME 2024 | AIME 2025 | MATH 500 | AMC 2023 | Minerva | O-Bench | Avg. |
|---|---|---|---|---|---|---|---|---|
| DS-Distill-Llama-8B | 32k+ / 32k | 0.506 | 0.346 | 0.896 | 0.815 | 0.295 | 0.541 | 0.566 |
| DS-Distill-Llama-8B | 32k+ / 8k | 0.348 | 0.238 | 0.825 | 0.652 | 0.267 | 0.440 | 0.462 |
| GRPO | 8k / 8k | 0.410 | 0.240 | 0.873 | 0.759 | 0.307 | 0.506 | 0.516 |
| GRPO+ER | 8k / 8k | 0.408 | 0.277 | 0.882 | 0.785 | 0.311 | 0.511 | 0.529 |
| Dr. GRPO | 8k / 8k | 0.423 | 0.285 | 0.867 | 0.786 | 0.300 | 0.497 | 0.526 |
| DAPO | 8k / 8k | 0.333 | 0.308 | 0.879 | 0.794 | 0.325 | 0.522 | 0.527 |
| TRPA | 8k / 8k | 0.454 | 0.279 | 0.864 | 0.756 | 0.289 | 0.518 | 0.527 |
| DisCO (L-ratio) | 8k / 8k | 0.506 | **0.356** | **0.900** | 0.831 | 0.326 | 0.553 | 0.579 |
| DisCO (log-L) | 8k / 8k | **0.523** | 0.354 | 0.896 | **0.843** | **0.331** | **0.560** | **0.584** |

**Models.** We conduct experiments with fine-tuning three models: DeepSeek-R1-Distill-Qwen-1.5B model (Q1.5B), DeepSeek-R1-Distill-Qwen-7B model (Q7B), and DeepSeek-R1-Distill-Llama-8B (L8B). All are distilled reasoning models.

**Baselines.** We primarily compare our methods with five most recent state-of-the-art reinforcement learning methods, including (1) GRPO [23]; (2) GRPO with an entropy regularization (GRPO-ER) that adds an entropy on probabilities of output tokens as a regularization to prevent entropy collapse, which is used by DeepScaleR [42]; (3) Dr. GRPO [40]; (4) DAPO's objective [79]; (5) TRPA [61]. For a comprehensive evaluation, we also include a set of reasoning models that are trained from the same base model by other studies with various techniques, such as (6) STILL-3-1.5B-preview [12], which adapt GRPO by periodically replacing the reference model after a fixed number of training steps; (7) DeepScaleR(DSR)-1.5B-Preview that uses maximum response length of 24k for training [42]; (8) GRPO-LEAD-7B [82], which extends GRPO by incorporating length-dependent

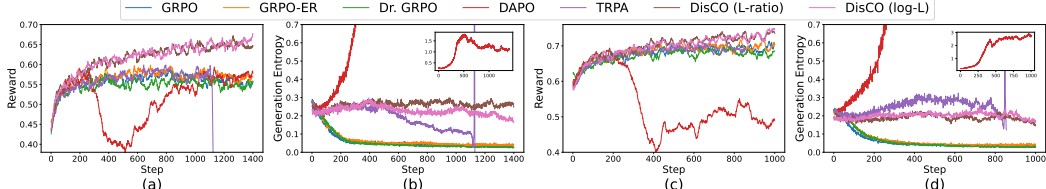

Figure 2: Training dynamics of different methods: left two are for fine-tuning DeepSeek-R1-Distill-Qwen-1.5B model and right two are for fine-tuning DeepSeek-R1-Distill-Qwen-7B model. (a), (c) plot the training reward (averaged over generated outputs for questions used in each step) vs the number of training steps (cf. Algorithm 1); (b), (d) plot the generation entropy vs training steps.

rewards, explicit penalty terms, and difficulty-based advantage reweighting to encourage concise and precise reasoning.

**Training Details.** For all the methods, we tune the constant learning rate in $[5e^{-7}, 1e^{-6}, 2e^{-6}]$ with AdamW optimizer with weight decay as 0.01. Generally, a learning rate of $2e^{-6}$ works better for the Q1.5B model, $1e^{-6}$ for the Q7B model, and $5e^{-7}$ for the L8B model. We employ a training batch size of 128, a mini-batch size of 32, and 8 responses for each question. The temperature is set to 0.6 for both training and evaluation, following the usage recommendation from [23]. For GRPO, $\beta$ is set to 0.001 as commonly used [12, 42]. For GRPO-ER, we use a coefficient of 0.001 for the entropy regularization [42]. For DAPO, we set $\epsilon_{low}$ to 0.2 and $\epsilon_{high}$ to 0.28 by following their paper. For our method, $\delta$ is set to $10^{-4}$ based on the empirical observation that the average KL divergence is around $2 * 10^{-5}$ and $\beta$ is set to $10^{3}$ such that the effective weight of the KL regularization when the constraint is violated by $\delta$ is on the order of $\beta * \delta = 0.1$. Since L-ratio and log-L scoring functions have different orders, we choose $\tau = 1$ for L-ratio and $\tau = 10$ for log-L scoring function, from $\{0.5, 1, 5, 10\}$. For fair comparisons, we do not implement Dynamic Sampling [79] for DAPO and other methods, as it introduces approximately three times the sampling cost at each training step. All methods are run for 1,400 steps on Q1.5B models and 1,000 steps on Q7B/L8B models. Evaluations are conducted every 200 steps, and the best performance for each method is reported.

## 6.1 Comparison with Baselines

**Performance.** We evaluate all the models across six mathematics-focused benchmark datasets to demonstrate the effectiveness of DisCO. The results are summarized in Table 2, 3 and 4. From Table 2 for Q1.5B models, we can observe that our proposed DisCO methods consistently outperform other baselines by a large margin. Notably, DisCO (log-L) with 8k length for both training and inference achieves an 7% average improvement over GRPO and surpasses DeepScaleR-1.5B-Preview that was trained with maximum 24k length and evaluated with 32k length. A similar trend is observed for Q7B models and L8B models (Table 3 and Table 4), where DisCO significantly outperforms all competing approaches.

**Training Dynamics.** We compare the training dynamics of different methods in terms of training rewards and generation entropy. From Figure 2 for fine-tuning Qwen-1.5B and Qwen-7B models, we can see that all baselines suffer from premature saturation due to either entropy collapse for GRPO, GRPO-ER, Dr. GRPO or excessive entropy growth of DAPO, which leads to an early deterministic or highly random policy. The entropy collapse phenomenon is also observed by [79, 27, 42]. TRPA that uses a KL divergence regularization is also observed with instability in the generation entropy in later steps (around 1100 for Q1.5B model and around 800 for Q7B model). In contrast, our methods with the two scoring functions are most stable, with training rewards kept increasing and generation entropy maintained around 0.22. We also include training dynamics for the L8B model in Appendix A.2, which follows a similar trend.

## 6.2 Ablation Studies

**DisCO vs DisCO-b.** Figure 3 (left) compares DisCO with DisCO-b using the L-ratio scoring function for training Q7B models for 1000 steps. The comparison clearly demonstrates the significant improvements of DisCO over DisCO-b, especially on the difficult AIME datasets. We also compare DisCO with DisCO-b for other settings in Appendix A.1, and observe that DisCO is consistently better than DisCO-b on average in all settings. It is also notable that DisCO-b with different scoring functions are also better than other baselines trained or evaluated by us in Table 2 and Table 3.

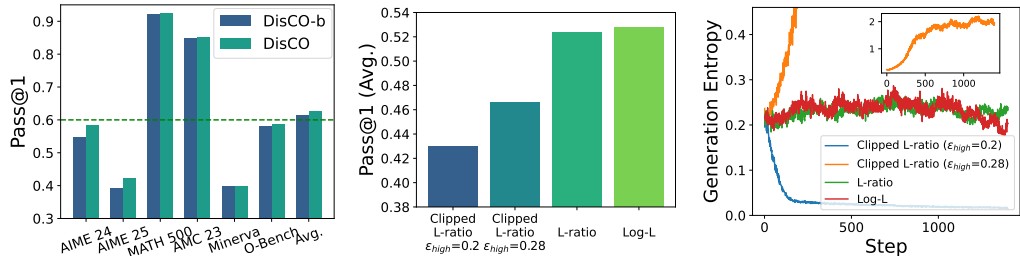

Figure 3: Ablation studies: left for comparing DisCO vs DisCO-b; middle and right for comparing clipping with non-clipping scoring functions.

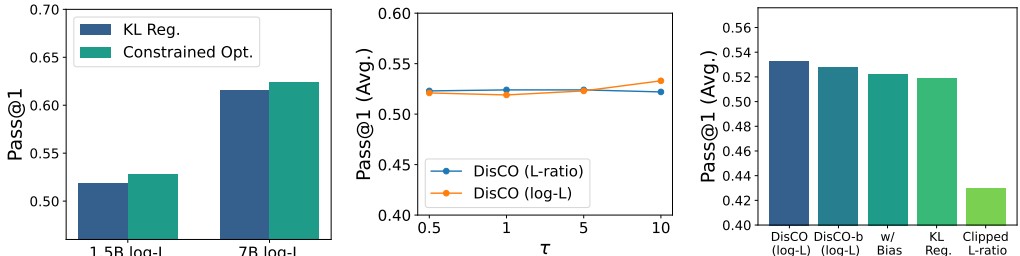

Figure 4: Ablation studies: left for comparing KL regularization vs constrained optimization; middle for sensitivity of DisCO w.r.t. the hyperparameter $\tau$; right for contribution of each component.

**Clipping vs Non-Clipping scoring functions.** We compare non-clipping scoring functions L-ratio, log-L with clipped L-ratio (5) in our DisCO-b approach for training Q1.5B models in Figure 3 (middle and right). For the clipped L-ratio, we adopt two versions: one with $\epsilon_{high} = 0.2$ to align with GRPO objective, and another with $\epsilon_{high} = 0.28$ similar to DAPO objective. We can see that clipped L-ratio with $\epsilon_{high} = 0.2$ causes entropy collapse while clipped L-ratio with $\epsilon_{high} = 0.28$ leads to excessively high entropy level, both yielding worse performance than non-clipping scoring functions.

**KL Regularization vs Constrained Optimization.** We investigate the advantages of constrained optimization over KL regularization for DisCO. Specifically, the KL regularization weight is set to the commonly used 0.001 [12, 42]. As shown in Figure 4 (left), constrained optimization performs better than KL regularization on both Q1.5B and Q7B models. Moreover, during our experiments, we observed that KL regularization leads to instability in training on Q7B models, similar to TRPA, which indicates that KL regularization is not sufficient to stabilize training.

**Sensitivity of hyperparameter $\tau$.** We study the sensitivity of DisCO to hyperparameter $\tau$ on training Q1.5B models. Similar to above experiments, we run DisCO for 1400 steps with different $\tau \in \{0.5, 1, 5, 10\}$. The result shown in Figure 3 (right) indicates that DisCO is not sensitive to $\tau$ in these ranges.

**Effect of each design choice.** We analyze the individual contribution of each component in DisCO by replacing its components separately with other designs. We experiment on Q1.5B models and compare with (1) DisCO-b that removes hard negative weighting; (2) adding question-level weight bias $\sqrt{p(q)(1 - p(q))}$ to DisCO-b, (3) replacing the KL-divergence constraint with a KL-divergence regularization in DisCO-b, and (4) using a clipping scoring function with $\epsilon_{high} = 0.2$ in DisCO-b, respectively. From Figure 4 (right), we can see that each of our proposed components is important in DisCO's improvement, where the use of a non-clipping scoring function is of vital importance.

## 7 Conclusion

In this work, we have proposed a novel discriminative constrained optimization framework for reinforcing large reasoning models, motivated by the analysis of the GRPO objective. The proposed framework is grounded in the principle of discriminative learning, avoiding difficulty bias and enhancing training stability with constrained trust region optimization. The experiments on mathematical reasoning demonstrated the significant superiority of our approaches, compared with GRPO and its recent variants. While this work focuses on binary rewards, future extensions could incorporate discriminative ranking objectives, like [9], to handle non-binary rewards. It would be interesting to apply the proposed approaches for fine-tuning larger models or other reasoning tasks.

## Acknowledgements

We are grateful for the reviewers' constructive comments. G. Li and T. Yang were partially supported by NSF grant #2306572.

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

# A    More Experimental Results

For all the experiments on 1.5B models, each run consumes 4*2 40G A100 GPUs and each training step takes approximately 6 minutes. For all the experiments on 7B models, each run consumes 1*8 80G H100 GPUs and each training step takes approximately 6.5 minutes.

## A.1    Detailed comparison between DisCO and DisCO-b

In this part, we compare DisCO and DisCO-b with different score functions on different models. As shown in Figure 5, DisCO consistently demonstrates better performance compared to DisCO-b across all settings, with higher average scores observed in each case. This consistent advantage highlights the effectiveness of the full DisCO framework. Additionally, it is worth emphasizing that even the DisCO-b variants, with L-ratio or log-L scoring functions, outperform all other baseline methods that are presented in Table 2 and Table 3. These results collectively underscore the robustness and general effectiveness of the DisCO approach.

## A.2    Training dynamics for fine-tuning 8B model.

In this part, we present the training dynamics of different methods for fine-tuning the DeepSeek-R1-Distill-Llama-8B model in Figure 6. Similar to observation in Figure 2 for fine-tuning 1.5B and 7B models, we can see that GRPO, GRPO-ER, and Dr. GRPO still suffer from entropy collapse while DAPO leads to excessive entropy growth, all accompanied by premature saturation in training reward. TRPA with a KL divergence regularization is also observed with instability in the training, indicating the insufficiency of KL regularization to stabilize training. In contrast, our methods with the two scoring functions and the KL constraint demonstrate the greatest stability, with training rewards continuing to rise and generation entropy remaining around 0.2.

## A.3    Experiments on DAPO-Math-17K dataset.

In order to demonstrate that the improvements achieved by DisCO are fundamental, rather than relying on specific properties of the dataset, we conducted additional experiments on the DAPO-Math-17K dataset [79] using 1.5B models, training them for 1400 steps. As shown in Table 5, DisCO

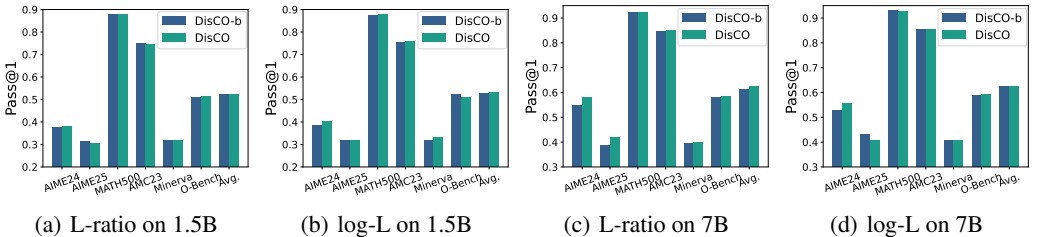

|  (a) L-ratio on 1.5B  |  (b) log-L on 1.5B  |  (c) L-ratio on 7B  |  (d) log-L on 7B  |

Figure 5: Comparison between DisCO-b and DisCO on different models with different score functions.

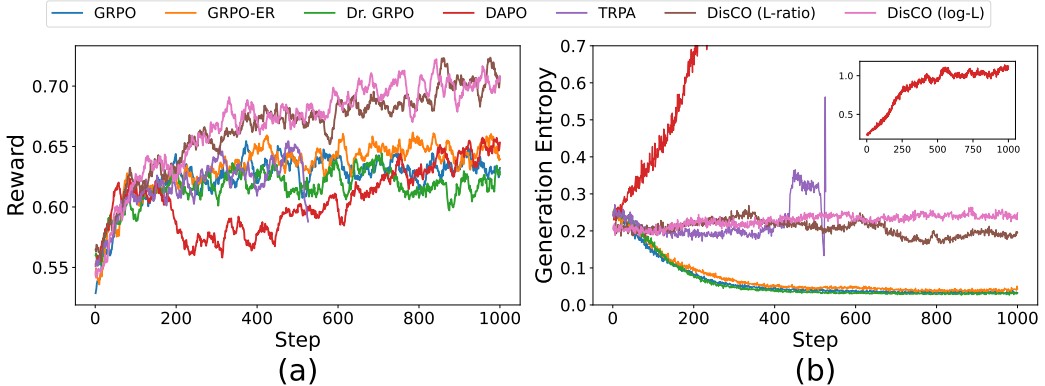

Figure 6: Training dynamics of different methods for fine-tuning DeepSeek-R1-Distill-Llama-8B model. (a) plots the training reward (averaged over generated outputs for questions used in each step) vs the number of training steps; (b) plots the generation entropy vs training steps.

Table 5: Comparison with baseline methods for fine-tuning DeepSeek-R1-Distill-Qwen-1.5B models on DAPO-Math-17K dataset.

| Model/Method | MRL(Train/Test) | AIME 2024 | AIME 2025 | MATH 500 | AMC 2023 | Minerva | O-Bench | Avg. |
|---|---|---|---|---|---|---|---|---|
| DS-Distill-Qwen-1.5B | 32k+ / 32k | 0.288 | 0.263 | 0.828 | 0.629 | 0.265 | 0.433 | 0.451 |
| DS-Distill-Qwen-1.5B | 32k+ / 8k | 0.181 | 0.215 | 0.758 | 0.515 | 0.237 | 0.353 | 0.376 |
| GRPO | 8k / 8k | 0.342 | 0.256 | 0.842 | 0.672 | 0.267 | 0.458 | 0.473 |
| GRPO-ER | 8k / 8k | 0.290 | 0.260 | 0.852 | 0.681 | 0.287 | 0.463 | 0.472 |
| Dr. GRPO | 8k / 8k | 0.300 | 0.250 | 0.849 | 0.705 | 0.292 | 0.464 | 0.477 |
| DAPO | 8k / 8k | 0.275 | 0.229 | 0.812 | 0.653 | 0.256 | 0.441 | 0.444 |
| TRPA | 8k / 8k | 0.346 | 0.279 | 0.836 | 0.683 | 0.281 | 0.450 | 0.479 |
| DisCO (L-ratio) | 8k / 8k | 0.413 | 0.310 | **0.874** | **0.775** | 0.307 | 0.495 | 0.529 |
| DisCO (log-L) | 8k / 8k | **0.460** | **0.317** | 0.873 | **0.775** | **0.320** | **0.502** | **0.541** |

methods still outperform other baselines by a large margin, demonstrating the generalizability of the proposed method to other datasets.

# B  More Theoretical Results

## B.1  Proof of Proposition 1

**Proof** *Since* $\mathbb{E}_{o\sim\pi_{old}(\cdot|q)}r(o|q) = p(q)$, $Var_{o\sim\pi_{old}(\cdot|q)}r(o|q) = p(q)(1-p(q))$, *we have*

$$A(o|q) = \begin{cases} \sqrt{\frac{1-p(q)}{p(q)}}, & \text{if } r(o|q) = 1, \\ -\sqrt{\frac{p(q)}{1-p(q)}}, & \text{if } r(o|q) = 0 \end{cases} \quad (12)$$

*According to (1), we have*

$$\mathbb{E}_q\mathbb{E}_{o\sim\pi_{old}(\cdot|q)}\left[\frac{1}{|o|}\sum_{t=1}^{|o|}f\left(\frac{\pi_\theta(o_t|q,o_{<t})}{\pi_{old}(o_t|q,o_{<t})}, A(o|q)\right)\right]$$

$$= \mathbb{E}_q\left[p(q)\mathbb{E}_{o\sim\pi_{old}^+(\cdot|q)}\frac{1}{|o|}\sum_{t=1}^{|o|}f\left(\frac{\pi_\theta(o_t|q,o_{<t})}{\pi_{old}(o_t|q,o_{<t})}, A(o|q)\right)\right.$$

$$\left. + (1-p(q))\mathbb{E}_{o\sim\pi_{old}^-(\cdot|q)}\frac{1}{|o|}\sum_{t=1}^{|o|}f\left(\frac{\pi_\theta(o_t|q,o_{<t})}{\pi_{old}(o_t|q,o_{<t})}, A(o|q)\right)\right]$$

$$= \mathbb{E}_q\left[p(q)\mathbb{E}_{o\sim\pi_{old}^+(\cdot|q)}\frac{1}{|o|}\sum_{t=1}^{|o|}f\left(\frac{\pi_\theta(o_t|q,o_{<t})}{\pi_{old}(o_t|q,o_{<t})}, \sqrt{\frac{1-p(q)}{p(q)}}\right)\right. \quad (13)$$

$$\left. + (1-p(q))\mathbb{E}_{o\sim\pi_{old}^-(\cdot|q)}\frac{1}{|o|}\sum_{t=1}^{|o|}f\left(\frac{\pi_\theta(o_t|q,o_{<t})}{\pi_{old}(o_t|q,o_{<t})}, -\sqrt{\frac{p(q)}{1-p(q)}}\right)\right]$$

$$= \mathbb{E}_q\sqrt{p(q)(1-p(q))}\left[\mathbb{E}_{o\sim\pi_{old}^+(\cdot|q)}\frac{1}{|o|}\sum_{t=1}^{|o|}f^+(\frac{\pi_\theta(o_t|q,o_{<t})}{\pi_{old}(o_t|q,o_{<t})}, 1)\right.$$

$$\left. - \mathbb{E}_{o\sim\pi_{old}^-(\cdot|q)}\frac{1}{|o|}\sum_{t=1}^{|o|}f^-(\frac{\pi_\theta(o_t|q,o_{<t})}{\pi_{old}(o_t|q,o_{<t})}, 1)\right]$$

*where the last equality is due to the assumption about $f(x,y)$. For GPRO, we have $f^+(x,1) = \min(x, clip(x, 1-\epsilon, 1+\epsilon)) = \min(x, 1+\epsilon)$ and $f^-(x,1) = \max(x, clip(x, 1-\epsilon, 1+\epsilon)) = \max(x, 1-\epsilon)$.*

---

**Algorithm 1** Discriminative Constrained Optimization

---

1: **Input:** Initial policy model $\pi_0$, reward function $r$, question set $\mathcal{D}$, hyperparameter $\delta, \beta, \tau$.
2: Policy model $\pi_\theta = \pi_0$
3: **for** Step $= 1, \cdots, T$ **do**
4:     Sample a batch of questions $\mathcal{B}$ from $\mathcal{D}$
5:     Update the old policy model $\pi_{\text{old}} = \pi_\theta$
6:     For each question $q \in \mathcal{B}$, sample $n$ responses $\{o_i\}_{i=1}^n \sim \pi_{\text{old}}(\cdot|q)$ denoted by $S_q$ and partition it into $S_q^+$ and $S_q^-$ based on rewards $r(o_i|q) \in \{0, 1\}$
7:     **for** minibatch $\mathcal{B}_m \in \mathcal{B}$ **do**
8:         Compute KL divergence estimator by

$$\hat{\mathbb{D}}_{KL} = \frac{1}{\sum_{q \in \mathcal{B}_m} \sum_{o \in S_q} |o|} \sum_{q \in \mathcal{B}_m} \sum_{o \in S_q} \sum_{t=1}^{|o|} \log \frac{\pi_{\text{old}}(o_t|q, o_{<t})}{\pi_\theta(o_t|q, o_{<t})}$$

9:         Compute gradient estimator of $\mathcal{J}_2(\theta)$ by

$$G_1 = \frac{1}{|\mathcal{B}_m|} \sum_{q \in \mathcal{B}_m} \frac{1}{|S_q^+|} \sum_{o \in S_q^+} \left( \nabla s_\theta(o, q) - \nabla \left( \tau \log \sum_{o' \in S_q^-} \exp(\frac{s_\theta(o', q)}{\tau}) \right) \right)$$

10:        Compute gradient estimator of constraint by $G_2 = 2\beta[\hat{\mathbb{D}}_{KL} - \delta]_+ \nabla \hat{\mathbb{D}}_{KL}$
11:        Update $\pi_\theta$ with Adam-W using the gradient estimator $G = G_1 + G_2$
12:     **end for**
13: **end for**

---

## B.2 Connection between discriminative objectives and surrogate objectives in RL

The score function L-ratio is inspired by the same principle as the surrogate objective in TRPO [56]. TRPO aims to maximize the following objective subject to a constraint:

$$\max_\theta \mathbb{E}_q \mathbb{E}_{o \sim \pi_{\theta_{old}}(\cdot|q)} \frac{1}{|o|} \sum_{t=1}^{|o|} \frac{\pi_\theta(o_t|q, o_{<t})}{\pi_{\text{old}}(o_t|q, o_{<t})} A(o_t) \tag{14}$$

$$s.t. \quad \mathbb{D}_{\text{KL}}(\pi_{\text{old}}||\pi_\theta) \le \delta.$$

When we apply the advantage function (12) to the objective, we have

$$\mathbb{E}_q \mathbb{E}_{o \sim \pi_{\text{old}}(\cdot|q)} \left[ \frac{1}{|o|} \sum_{t=1}^{|o|} \frac{\pi_\theta(o_t|q, o_{<t})}{\pi_{\text{old}}(o_t|q, o_{<t})} A(o|q) \right]$$

$$= \mathbb{E}_q \Bigg[ p(q) \mathbb{E}_{o \sim \pi_{\text{old}}^+(\cdot|q)} \frac{1}{|o|} \sum_{t=1}^{|o|} \frac{\pi_\theta(o_t|q, o_{<t})}{\pi_{\text{old}}(o_t|q, o_{<t})} A(o|q)$$

$$+ (1 - p(q)) \mathbb{E}_{o \sim \pi_{\text{old}}^-(\cdot|q)} \frac{1}{|o|} \sum_{t=1}^{|o|} \frac{\pi_\theta(o_t|q, o_{<t})}{\pi_{\text{old}}(o_t|q, o_{<t})} A(o|q) \Bigg]$$

$$= \mathbb{E}_q \Bigg[ p(q) \mathbb{E}_{o \sim \pi_{\text{old}}^+(\cdot|q)} \frac{1}{|o|} \sum_{t=1}^{|o|} \frac{\pi_\theta(o_t|q, o_{<t})}{\pi_{\text{old}}(o_t|q, o_{<t})} * \sqrt{\frac{1 - p(q)}{p(q)}}$$

$$+ (1 - p(q)) \mathbb{E}_{o \sim \pi_{\text{old}}^-(\cdot|q)} \frac{1}{|o|} \sum_{t=1}^{|o|} \frac{\pi_\theta(o_t|q, o_{<t})}{\pi_{\text{old}}(o_t|q, o_{<t})} * (-\sqrt{\frac{p(q)}{1 - p(q)}}) \Bigg]$$

$$= \mathbb{E}_q \sqrt{p(q)(1 - p(q))} \Bigg[ \mathbb{E}_{o \sim \pi_{\text{old}}^+(\cdot|q)} \frac{1}{|o|} \sum_{t=1}^{|o|} \frac{\pi_\theta(o_t|q, o_{<t})}{\pi_{\text{old}}(o_t|q, o_{<t})} - \mathbb{E}_{o \sim \pi_{\text{old}}^-(\cdot|q)} \frac{1}{|o|} \sum_{t=1}^{|o|} \frac{\pi_\theta(o_t|q, o_{<t})}{\pi_{\text{old}}(o_t|q, o_{<t})} \Bigg]$$

$$\tag{15}$$

This gives the exact scoring function L-ratio: $s_\theta(o, q) = \frac{1}{|o|} \sum_{t=1}^{|o|} \frac{\pi_\theta(o_t|q, o_{<t})}{\pi_{\text{old}}(o_t|q, o_{<t})}$. After removing the improper weight $\sqrt{p(q)(1 - p(q))}$, Eqn. (15) is same as Eqn. (7) with $\ell(s) = s$.

In the reinforcement learning literature, vanilla policy gradient methods also gain significant attention due to their simplicity and remarkable performance. The vanilla policy gradient(VPG) methods work

Table 6: Weighted discriminative objectives and their scoring functions $s^+(o,q)$ and $s^-(o,q)$ for different methods, where $\sigma(\cdot)$ is the sigmoid function.

| Objective | $\mathbb{E}_q \omega(q) \mathbb{E}_{o \sim \pi_{\text{old}}^+(\cdot|q), o' \sim \pi_{\text{old}}^-(\cdot|q)} \ell\big(s_\theta^+(o,q) - s_\theta^-(o',q)\big)$ |
|---|---|
| GRPO | $\omega(q) = \sqrt{p(q)(1-p(q))}, \quad \ell(s) = s$ |
| | $s_\theta^+(o,q) = \frac{1}{|o|}\sum_{t=1}^{|o|} \min(\frac{\pi_\theta(o_t|q,o_{<t})}{\pi_{\text{old}}(o_t|q,o_{<t})}, 1+\epsilon), \quad s_\theta^-(o',q) = \frac{1}{|o'|}\sum_{t=1}^{|o|} \max(\frac{\pi_\theta(o'_t|q,o'_{<t})}{\pi_{\text{old}}(o'_t|q,o'_{<t})}, 1-\epsilon)$ |
| Dr. GRPO | $\omega(q) = p(q)(1-p(q)), \quad \ell(s) = s$ |
| | $s_\theta^+(o,q) = \sum_{t=1}^{|o|} \min(\frac{\pi_\theta(o_t|q,o_{<t})}{\pi_{\text{old}}(o_t|q,o_{<t})}, 1+\epsilon), \quad s_\theta^-(o',q) = \sum_{t=1}^{|o'|} \max(\frac{\pi_\theta(o'_t|q,o'_{<t})}{\pi_{\text{old}}(o'_t|q,o'_{<t})}, 1-\epsilon)$ |
| DAPO | $\omega(q) = \sqrt{p(q)(1-p(q))}, \quad \ell(s) = s$ |
| | $s_\theta^+(o,q) = \frac{1}{\mathbb{E}_{o \sim \pi_{\text{old}}(\cdot|q)}|o|}\sum_{t=1}^{|o|} \min(\frac{\pi_\theta(o_t|q,o_{<t})}{\pi_{\text{old}}(o_t|q,o_{<t})}, 1+\epsilon_{high}), \quad s_\theta^-(o',q) = \frac{1}{\mathbb{E}_{o \sim \pi_{\text{old}}(\cdot|q)}|o|}\sum_{t=1}^{|o'|} \max(\frac{\pi_\theta(o'_t|q,o'_{<t})}{\pi_{\text{old}}(o'_t|q,o'_{<t})}, (1-\epsilon_{low}))$ |
| GPG | $\omega(q) = \alpha p(q)(1-p(q)), \quad \ell(s) = s$ |
| | $s_\theta^+(o,q) = \frac{1}{\mathbb{E}_{o \sim \pi_{\text{old}}(\cdot|q)}|o|}\sum_{t=1}^{|o|} \log \pi_\theta(o_t|q,o_{<t}), \quad s_\theta^-(o',q) = \frac{1}{\mathbb{E}_{o \sim \pi_{\text{old}}(\cdot|q)}|o|}\sum_{t=1}^{|o'|} \log \pi_\theta(o'_t|q,o'_{<t})$ |
| TRPA | $\omega(q) = 1, \quad \ell(s) = \log(\sigma(\beta(o)s))$ |
| | $s_\theta^+(o,q) = \sum_{t=1}^{|o|} \log \frac{\pi_\theta(o_t|q,o_{<t})}{\pi_{ref}(o_t|q,o_{<t})}, \quad s_\theta^-(o',q) = \sum_{t=1}^{|o'|} \log \frac{\pi_\theta(o'_t|q,o'_{<t})}{\pi_{ref}(o'_t|q,o'_{<t})}$ |

by computing an estimator of the policy gradient and plugging it into a stochastic gradient algorithm. The most commonly used surrogate objective function for gradient estimator has the form:

$$\mathcal{J}_{\text{VPG}} = \mathbb{E}_q \mathbb{E}_{o \sim \pi_{\theta_{old}}(\cdot|q)} \frac{1}{|o|} \sum_{t=1}^{|o|} \log \pi_\theta(o_t|q, o_{<t}) A(o_t) \tag{16}$$

Similar to the derivation above, by plugging in the advantage estimator Eqn. (12), we have:

$$\mathcal{J}_{\text{VPG}} = \mathbb{E}_q \sqrt{p(q)(1-p(q))} \bigg[ \mathbb{E}_{o \sim \pi_{\text{old}}^+(\cdot|q)} \frac{1}{|o|} \sum_{t=1}^{|o|} \log \pi_\theta(o_t|q, o_{<t})$$
$$- \mathbb{E}_{o \sim \pi_{\text{old}}^-(\cdot|q)} \frac{1}{|o|} \sum_{t=1}^{|o|} \log \pi_\theta(o_t|q, o_{<t}) \bigg] \tag{17}$$

This directly motivate the score function log-L: $s_\theta(o,q) = \frac{1}{|o|}\sum_{t=1}^{|o|} \log \pi_\theta(o_t|q,o_{<t})$. After removing the inappropriate weight on questions, Eqn. (17) is same as Eqn. (7) with $\ell(s) = s$.

### B.3 Analysis of other variants of GRPO

In this part, we show that the other variants of GRPO still have difficulty bias on questions.

Let's start with Dr. GRPO. In Dr. GRPO, the un-normalized advantage function is employed:

$$\hat{A}(o|q) = \begin{cases} 1 - p(q), & \text{if } r(o|q) = 1, \\ -p(q), & \text{if } r(o|q) = 0 \end{cases} \tag{18}$$

With $f(x,y) = \min(xy, \text{clip}(x, 1-\epsilon, 1+\epsilon)y)$, we have

$$\mathcal{J}_{\text{Dr.GRPO}} = \mathbb{E}_q \mathbb{E}_{o \sim \pi_{\text{old}}(\cdot|q)} \bigg[ \sum_{t=1}^{|o|} f\bigg( \frac{\pi_\theta(o_t|q,o_{<t})}{\pi_{\text{old}}(o_t|q,o_{<t})}, \hat{A}(o|q) \bigg) \bigg]$$

$$= \mathbb{E}_q \bigg[ p(q) \mathbb{E}_{o \sim \pi_{\text{old}}^+(\cdot|q)} \sum_{t=1}^{|o|} f\bigg( \frac{\pi_\theta(o_t|q,o_{<t})}{\pi_{\text{old}}(o_t|q,o_{<t})}, \hat{A}(o|q) \bigg)$$

$$+ (1-p(q)) \mathbb{E}_{o \sim \pi_{\text{old}}^-(\cdot|q)} \sum_{t=1}^{|o|} f\bigg( \frac{\pi_\theta(o_t|q,o_{<t})}{\pi_{\text{old}}(o_t|q,o_{<t})}, \hat{A}(o|q) \bigg) \bigg] \tag{19}$$

$$= \mathbb{E}_q \bigg[ p(q) \mathbb{E}_{o \sim \pi_{\text{old}}^+(\cdot|q)} \sum_{t=1}^{|o|} f\bigg( \frac{\pi_\theta(o_t|q,o_{<t})}{\pi_{\text{old}}(o_t|q,o_{<t})}, 1-p(q) \bigg)$$

$$+ (1-p(q)) \mathbb{E}_{o \sim \pi_{\text{old}}^-(\cdot|q)} \sum_{t=1}^{|o|} f\bigg( \frac{\pi_\theta(o_t|q,o_{<t})}{\pi_{\text{old}}(o_t|q,o_{<t})}, -p(q) \bigg) \bigg]$$

$$= \mathbb{E}_q p(q)(1-p(q)) \bigg[ \mathbb{E}_{o \sim \pi_{\text{old}}^+(\cdot|q)} s_\theta^+(o,q) - \mathbb{E}_{o \sim \pi_{\text{old}}^-(\cdot|q)} s_\theta^-(o,q) \bigg]$$

where $s_\theta^+(o, q) = \sum_{t=1}^{|o|} \min(\frac{\pi_\theta(o_t|q, o_{<t})}{\pi_{\text{old}}(o_t|q, o_{<t})}, 1 + \epsilon)$, $s_\theta^-(o, q) = \sum_{t=1}^{|o|} \max(\frac{\pi_\theta(o_t|q, o_{<t})}{\pi_{\text{old}}(o_t|q, o_{<t})}, 1 - \epsilon)$. We can see that the difficult bias $\omega(q) = p(q)(1 - p(q))$ on questions persists in Dr. GRPO.

Secondly, let's reformulate the DAPO objective to show the question-level bias. With $f(x, y) = \min(xy, \text{clip}(x, 1 - \epsilon_{low}, 1 + \epsilon_{high})y)$ and advantage estimator (12), the expected version of DAPO is

$$
\begin{aligned}
\mathcal{J}_{\text{DAPO}} &= \mathbb{E}_q \mathbb{E}_{o \sim \pi_{\text{old}}(\cdot|q)} \left[ \frac{1}{\mathbb{E}_{o \sim \pi_{\text{old}}(\cdot|q)}|o|} \sum_{t=1}^{|o|} f\left( \frac{\pi_\theta(o_t|q, o_{<t})}{\pi_{\text{old}}(o_t|q, o_{<t})}, A(o|q) \right) \right] \\
&= \mathbb{E}_q \left[ p(q) \mathbb{E}_{o \sim \pi_{\text{old}}^+(\cdot|q)} \frac{1}{\mathbb{E}_{o \sim \pi_{\text{old}}(\cdot|q)}|o|} \sum_{t=1}^{|o|} f\left( \frac{\pi_\theta(o_t|q, o_{<t})}{\pi_{\text{old}}(o_t|q, o_{<t})}, A(o|q) \right) \right. \\
&\quad \left. + (1 - p(q)) \mathbb{E}_{o \sim \pi_{\text{old}}^-(\cdot|q)} \frac{1}{\mathbb{E}_{o \sim \pi_{\text{old}}(\cdot|q)}|o|} \sum_{t=1}^{|o|} f\left( \frac{\pi_\theta(o_t|q, o_{<t})}{\pi_{\text{old}}(o_t|q, o_{<t})}, A(o|q) \right) \right] \\
&= \mathbb{E}_q \left[ p(q) \mathbb{E}_{o \sim \pi_{\text{old}}^+(\cdot|q)} \frac{1}{\mathbb{E}_{o \sim \pi_{\text{old}}(\cdot|q)}|o|} \sum_{t=1}^{|o|} f\left( \frac{\pi_\theta(o_t|q, o_{<t})}{\pi_{\text{old}}(o_t|q, o_{<t})}, \sqrt{\frac{1 - p(q)}{p(q)}} \right) \right. \\
&\quad \left. + (1 - p(q)) \mathbb{E}_{o \sim \pi_{\text{old}}^-(\cdot|q)} \frac{1}{\mathbb{E}_{o \sim \pi_{\text{old}}(\cdot|q)}|o|} \sum_{t=1}^{|o|} f\left( \frac{\pi_\theta(o_t|q, o_{<t})}{\pi_{\text{old}}(o_t|q, o_{<t})}, -\sqrt{\frac{p(q)}{1 - p(q)}} \right) \right] \\
&= \mathbb{E}_q \sqrt{p(q)(1 - p(q))} \left[ \mathbb{E}_{o \sim \pi_{\text{old}}^+(\cdot|q)} s_\theta^+(o, q) - \mathbb{E}_{o \sim \pi_{\text{old}}^-(\cdot|q)} s_\theta^-(o, q) \right]
\end{aligned}
$$

(20)

where $s_\theta^+(o, q) = \frac{1}{\mathbb{E}_{o \sim \pi_{\text{old}}(\cdot|q)}|o|} \sum_{t=1}^{|o|} \min(\frac{\pi_\theta(o_t|q, o_{<t})}{\pi_{\text{old}}(o_t|q, o_{<t})}, 1 + \epsilon)$, $s_\theta^-(o, q) = \frac{1}{\mathbb{E}_{o \sim \pi_{\text{old}}(\cdot|q)}|o|} \sum_{t=1}^{|o|} \max(\frac{\pi_\theta(o_t|q, o_{<t})}{\pi_{\text{old}}(o_t|q, o_{<t})}, 1 - \epsilon)$. We can see that the difficult bias $\omega(q) = \sqrt{p(q)(1 - p(q))}$ is placed on questions persists in DAPO.

Thirdly, we show the difficult bias in GPG objective. In GPG, $\alpha \hat{A}(o|q)$ is employed as their advantage estimator. Thus, we have

$$
\begin{aligned}
\mathcal{J}_{\text{GPG}} &= \mathbb{E}_q \mathbb{E}_{o \sim \pi_{\text{old}}(\cdot|q)} \left[ \frac{1}{\mathbb{E}_{o \sim \pi_{\text{old}}(\cdot|q)}|o|} \sum_{t=1}^{|o|} \alpha \log \pi_\theta(o_t|q, o_{<t}) \hat{A}(o|q) \right] \\
&= \mathbb{E}_q \left[ p(q) \mathbb{E}_{o \sim \pi_{\text{old}}^+(\cdot|q)} \frac{1}{\mathbb{E}_{o \sim \pi_{\text{old}}(\cdot|q)}|o|} \sum_{t=1}^{|o|} \alpha \log \pi_\theta(o_t|q, o_{<t}) \hat{A}(o|q) \right. \\
&\quad \left. + (1 - p(q)) \mathbb{E}_{o \sim \pi_{\text{old}}^-(\cdot|q)} \frac{1}{\mathbb{E}_{o \sim \pi_{\text{old}}(\cdot|q)}|o|} \sum_{t=1}^{|o|} \alpha \log \pi_\theta(o_t|q, o_{<t}) \hat{A}(o|q) \right] \\
&= \mathbb{E}_q \left[ p(q) \mathbb{E}_{o \sim \pi_{\text{old}}^+(\cdot|q)} \frac{1}{\mathbb{E}_{o \sim \pi_{\text{old}}(\cdot|q)}|o|} \sum_{t=1}^{|o|} \alpha \log \pi_\theta(o_t|q, o_{<t}) * (1 - p(q)) \right. \\
&\quad \left. + (1 - p(q)) \mathbb{E}_{o \sim \pi_{\text{old}}^-(\cdot|q)} \frac{1}{\mathbb{E}_{o \sim \pi_{\text{old}}(\cdot|q)}|o|} \sum_{t=1}^{|o|} \alpha \log \pi_\theta(o_t|q, o_{<t}) * (-p(q)) \right] \\
&= \mathbb{E}_q \alpha p(q)(1 - p(q)) \left[ \mathbb{E}_{o \sim \pi_{\text{old}}^+(\cdot|q)} s_\theta^+(o, q) - \mathbb{E}_{o \sim \pi_{\text{old}}^-(\cdot|q)} s_\theta^-(o, q) \right]
\end{aligned}
$$

(21)

where $s_\theta^+(o, q) = \frac{1}{\mathbb{E}_{o \sim \pi_{\text{old}}(\cdot|q)}|o|} \sum_{t=1}^{|o|} \log \pi_\theta(o_t|q, o_{<t})$, and $s_\theta^-(o, q) = \frac{1}{\mathbb{E}_{o \sim \pi_{\text{old}}(\cdot|q)}|o|} \sum_{t=1}^{|o|} \log \pi_\theta(o_t|q, o_{<t})$. We can see that the difficult bias $\omega(q) = \alpha p(q)(1 - p(q))$ is on questions in GPG.

Finally, we summarize the question-level weights and their score functions for other variants of GRPO in Table 6.

