# OpenReview forum: "DisCO: Reinforcing Large Reasoning Models with Discriminative Constrained Optimization"
_NeurIPS.cc/2025/Conference — NeurIPS 2025 poster_

### Official Review · Reviewer_ewTG · 2025-06-30

**Clarity:** 3
**Significance:** 3
**Originality:** 3
**Rating:** 5
**Confidence:** 3

**Summary:**

The authors propose a new reinforcement learning (RL) optimization framework for large reasoning models based on discriminative learning. Specifically, it employs a score-based discriminative objective instead of a group-relative policy and uses a scoring function in place of clipping. Training is stabilized through a simple constrained optimization method. The proposed framework demonstrates strong potential across multiple reasoning benchmarks, achieving better performance than GRPO while producing shorter responses.

**Questions:**

- What happens if we increase the response length (e.g., 24k or 32k)? Does it improve DisCO's performance notably?

**Ethical Concerns:**

["NO or VERY MINOR ethics concerns only"]

**Final Justification:**

The authors addressed my concerns. I will increase my score (4-->5).

**Limitations:**

yes

**Quality:**

3

**Strengths And Weaknesses:**

**[Strengths]**
- The paper is well-written and easy to follow.
- The proposed framework to alleviate difficulty bias is interesting and achieves meaningful results across multiple benchmarks.
- The ablation study is detailed and comprehensive.

**[Weaknesses]**
- The framework only considers binary rewards (though the authors acknowledge this). It would be more compelling if it were shown to work with non-binary reward settings as well.
- The paper misses an ablation study on response length. It would be helpful to understand how DisCO's performance varies with different response lengths.

---

> ### Author Rebuttal · Authors · 2025-07-30
>
> We thank the reviewer for dedicating the time to providing helpful feedback on our paper.
>
> **Q1:** The framework only considers binary rewards (though the authors acknowledge this). It would be more compelling if it were shown to work with non-binary reward settings as well.
>
> **A:** We agree that it will strengthen the work. However, the focus on the binary reward setting helps us to better illustrate the limitation of GRPO through theoretical analysis. We believe the improvement brought by DisCO is non-trivial and expect to extend its strength to non-binary rewards by considering other discriminative objectives, e.g., ranking losses.
>
> Additionally, to make our work more compelling, we further validated our method on a model from another LLM-family, i.e., DeepSeek-R1-Distill-Llama-8B, training them for 1000 steps. The results are summarized below. We observed that our proposed DisCO methods still  outperform other baselines by a large margin, demonstrating the generalizability of the proposed method to other models.
>
> | Model               | MRL(Train/Test) | AIME 2024 | AIME 2025 | MATH 500 | AMC 2023 | Minerva | O-Bench | Avg.  |
> | ------------------- | --------------- | --------- | --------- | -------- | -------- | ------- | ------- | ----- |
> | DS-Distill-Llama-8B | 32k+   /   32k  | 0.506     | 0.346     | 0.896    | 0.815    | 0.295   | 0.541   | 0.566 |
> | DS-Distill-Llama-8B | 32k+  /   8k    | 0.348     | 0.238     | 0.825    | 0.652    | 0.267   | 0.440   | 0.462 |
> | GRPO                | 8k   /    8k    | 0.410     | 0.240     | 0.873    | 0.759    | 0.307   | 0.506   | 0.516 |
> | GRPO-ER             | 8k   /   8k     | 0.408     | 0.277     | 0.882    | 0.785    | 0.311   | 0.511   | 0.529 |
> | Dr. GRPO            | 8k   /   8k     | 0.423     | 0.285     | 0.867    | 0.786    | 0.300   | 0.497   | 0.526 |
> | DAPO                | 8k   /   8k     | 0.333     | 0.308     | 0.879    | 0.794    | 0.325   | 0.522   | 0.527 |
> | TRPA                | 8k   /   8k     | 0.454     | 0.279     | 0.864    | 0.756    | 0.289   | 0.518   | 0.527 |
> | DisCO (L-ratio)     | 8k   /   8k     | 0.506     | **0.356**     | **0.900**    | 0.831    | 0.326   | 0.553   | 0.579 |
> | DisCO (log-L)       | 8k   /   8k     | **0.523**     | 0.354     | 0.896    | **0.843**    | **0.331**   | **0.560**   | **0.584** |
>
> **Q2:** It would be helpful to understand how DisCO's performance varies with different response lengths. For example, What happens if we increase the response length (e.g., 24k or 32k)?
>
> **A:** We thank the reviewer for the constructive comment. We have conducted an experiment using 16k response length for training and inference on 1.5B models. The results shown below demonstrate that increasing response length brings additional improvements, compared with 8k length for training and inference.
>
> | Model                | MRL(Train/Test) | AIME 2024 | AIME 2025 | MATH 500 | AMC 2023 | Minerva | O-Bench | Avg.  |
> | -------------------- | --------------- | --------- | --------- | -------- | -------- | ------- | ------- | ----- |
> | DS-Distill-Qwen-1.5B | 32k+   /   32k  | 0.288     | 0.263     | 0.828    | 0.629    | 0.265   | 0.433   | 0.451 |
> | DSR-1.5B-Preview     | 24k   /   32k   | 0.431     | 0.304     | 0.878    | 0.736    | 0.302   | 0.500   | 0.525 |
> | DisCO (L-ratio)      | 8k   /   8k     | 0.381     | 0.306     | 0.878    | 0.746    | 0.319   | 0.512   | 0.524 |
> | DisCO (log-L)        | 8k   /   8k     | 0.404     | 0.317     | 0.876    | 0.758    | 0.333   | 0.509   | 0.533 |
> | DisCO (L-ratio)      | 16k   /   16k   | 0.410     | 0.340     | 0.885    | 0.748    | 0.324   | 0.515   | 0.537 |
> | DisCO (log-L)        | 16k   /   16k   | 0.404     | 0.333     | 0.891    | 0.761    | 0.321   | 0.531   | 0.540 |

---

> > ### Author Response · Authors · 2025-08-05
> > **Hope to discuss with you!**
> >
> > Dear Reviewer ewTG,
> >
> > Thank you for your thoughtful review and encouraging rating!
> >
> > As the author-reviewer discussion period is nearing its end, we would like to follow up to see if our rebuttal has addressed your concerns. Please let us know if any further clarification would be helpful.
> >
> > Thank you again!
> >
> > Authors

---

> > ### Comment · Reviewer_ewTG · 2025-08-06
> >
> > Thanks for your response. The authors have addressed all my concerns.

---

### Official Review · Reviewer_CL8D · 2025-07-02

**Clarity:** 3
**Significance:** 3
**Originality:** 2
**Rating:** 5
**Confidence:** 3

**Summary:**

This paper addresses the problem difficulty bias and training instability in existing Large Reasoning Models (LRMs) during reinforcement learning by proposing a novel framework. The study optimizes scoring functions to enhance scores for correct answers while suppressing those for incorrect answers, employing a constrained optimization approach to ensure training stability. Additionally, it introduces Distributionally Robust Optimization (DRO) techniques to mitigate data imbalance issues during training. Experiments demonstrate that the proposed architecture outperforms existing GRPO and its variants on reasoning tasks, significantly enhancing model performance.

**Questions:**

1.Could you quantify the individual contribution rates of each improvement in DisCO through ablation studies in detail?
2.Will the DisCO algorithm pose computational efficiency challenges as models scale further?
3.In hyperparameter selection, have you accounted for the impact of different hyperparameter combinations on model generalization capabilities?

**Ethical Concerns:**

["NO or VERY MINOR ethics concerns only"]

**Final Justification:**

Thanks for the authors' thorough and clear responses. All my questions have been fully addressed, and I have no further comments. Therefore, I recommend acceptance.

**Limitations:**

Yes，the authors have explicitly stated several limitations of their work in the paper, noting that their method primarily
optimizes for binary rewards and that experiments were constrained by computational resources, covering only 1.5B and 7B models.

**Paper Formatting Concerns:**

None.

**Quality:**

3

**Strengths And Weaknesses:**

Strengths
1.The paper offers detailed theoretical analysis and experimental validation, exposes GRPO's limitations, and presents targeted solutions. The well-designed experiments cover multiple benchmark tasks and various model sizes. The results show that the new architecture in the paper is better than existing methods in improving model performance. Also, the paper discloses enough experimental details for other researchers to reproduce the experiments.
2.This study is highly significant for enhancing the reasoning abilities of large - scale reasoning models in complex tasks. By addressing key issues in current methods, it offers a more effective and stable model optimization approach, which will advance research in related fields.

Weaknesses
1.In the experimental part, although the paper provides a large number of experimental results and analyses, the descriptions of certain experimental details may not be in-depth enough, such as the selection and tuning process of hyperparameters.
2.When presenting related work and introducing the DisCO method, the paper emphasizes the limitations of GRPO and its variants but rarely mentions their merits and applicability in specific scenarios. This may bias readers' understanding of different methods, preventing them from objectively evaluating the relative advantages and disadvantages of each approach in various contexts.

---

> ### Author Rebuttal · Authors · 2025-07-30
>
> We thank the reviewer for providing helpful comments on improving our paper. Below we would like to answer raised questions.
>
> **Q1:** In hyperparameter selection, have you accounted for the impact of different hyperparameter combinations on model generalization capabilities?
>
>
> **A:** We tuned two hyper-parameters for DisCO, i.e., the learning rate and the $\tau$. The process of hyperparameter selection is the same as model selection, which accounts for the generalization performance. We report the performance of the best hyper-parameters in the tuning range that yields the best average test performance. We did the same model selection for all baselines. We will add more details in the revision.
>
>
> **Q2:** Mentioning the merits and applicability of GRPO in specific scenarios.
>
> A: We thank the reviewer for the constructive comment. In the opening paragraph of the Introduction, we highlighted the importance of GRPO in achieving performance comparable to advanced proprietary models across many reasoning benchmarks. Additionally, in the second paragraph of the Related Work section, we discussed key contributions of GRPO and its variants to provide readers with a foundational understanding of each approach.
>
> We agree with the reviewer that each method has its own strengths. In Section 7, we acknowledged the limitation of our work in focusing on binary rewards, whereas GRPO and its variants can naturally handle non-binary reward signals without modification. We will make this clearer in the revision.
>
>
>
> **Q3:** Could you quantify the individual contribution rates of each improvement in DisCO through ablation studies in detail?
>
> **A:** Thank you for the helpful suggestion. The proposed method incorporates three key design choices:
> (1) removing question-level weight bias,
> (2) using a non-clipping scoring function, and
> (3) applying a KL-divergence constraint.
>
> The effectiveness of (1) has been demonstrated in the comparison between GRPO and GRPO$_\text{rw}$, as shown in Fig. 1 (c, d). Additionally, we conducted new experiments on training a 1.5B model, comparing DisCO with its variant  that includes question-level weight bias while keeping all other components unchanged. The results are presented in the table below.
>
> The benefit of (2) is validated in Figure 3 (second from the left), which shows that using the non-clipping scoring functions yields significant improvements over its clipping-based counterpart, when all other components are the same as DisCO.
>
> To evaluate the impact of (3), we conducted another experiment on the 1.5B model, comparing DisCO with a variant that replaces the KL constraint with KL-divergence regularization. The results are also summarized in the table below.
>
> We can see that each component contributes to the improvements, where the use of non-clipping scoring function contributes the most. We will incorporate the discussion into our ablation studies.
>
> | Method                     | [pass@1(Avg.)](mailto:pass@1(Avg.)) | Method                     | [pass@1(Avg.)](mailto:pass@1(Avg.)) |
> | ------------------------- | -----------------------------------  | ------------------------- | ----------------------------------- |
> | DisCO (log-L)             | 0.533                                | DisCO (L-ratio)           | 0.524                               |
> | DisCO --> w/ weight bias            | 0.522 (↓ 0.011)            | DisCO --> w/ weight bias             | 0.511  (↓ 0.013)          |
> | DisCO -->Clipping scoring  func. | 0.430  (↓ 0.103)              | DisCO --> Clipping scoring func.| 0.430  (↓ 0.094)                |
> | DisCO--> KL Regularization          | 0.519    (↓ 0.014)                   | DisCO -->  KL Regularization         | 0.523 (↓ 0.001)             |
>
>
> **Q4:** Will the DisCO algorithm pose computational efficiency challenges as models scale further?
>
> **A:** The DisCO algorithm does **not** introduce any extra computational overhead, compared with GRPO and its variants. They have similar computational costs.

---

> > ### Author Response · Authors · 2025-08-05
> > **Hope to discuss with you!**
> >
> > Dear Reviewer CL8D,
> >
> > Thank you for your thoughtful review and encouraging rating!
> >
> > As the author-reviewer discussion period is nearing its end, we would like to follow up to see if our rebuttal has addressed your concerns. Please let us know if any further clarification would be helpful.
> >
> > Thank you again!
> >
> > Authors

---

> > ### Comment · Reviewer_CL8D · 2025-08-06
> >
> > Thank you for your thorough and clear responses. All my questions have been fully addressed, and I have no further comments. Therefore, I recommend acceptance.

---

### Official Review · Reviewer_Hi9A · 2025-07-03

**Clarity:** 2
**Significance:** 3
**Originality:** 3
**Rating:** 5
**Confidence:** 3

**Summary:**

The paper proposes a new Discriminative Constrained Optimization framework for reinforcing large reasoning models (LRMs). Starting from group relative policy optimization (GRPO), DisCO firstly replace the group relative objective with a discriminative objective defined by a scoring function, and employs a constrained optimization approach to enforce the KL divergence constrainet. Experimental results show that DisCO outperforms GRPO and DAPO on 1.5B and 7B model with the same training queries.

**Questions:**

1. The equation 9 seems to have very similar formulation as Direct Policy Optimization. Is there any essential connection and difference between the proposed DisCO and online DPO?
2. The reward distribution of training queries are highly correlated with the initial policy and the training data. If the initial policy is strong, and the training queries are hard (for example, using the queries provided by DAPO[1] or OREAL[2], is DisCO still effective and better than previous methods?
[1] DAPO: An Open-Source LLM Reinforcement Learning System at Scale https://huggingface.co/datasets/BytedTsinghua-SIA/DAPO-Math-17k
[2] OREAL: Exploring the Limit of Outcome Reward for Learning Mathematical Reasoning.  https://huggingface.co/datasets/internlm/OREAL-RL-Prompts

**Ethical Concerns:**

["NO or VERY MINOR ethics concerns only"]

**Final Justification:**

The authors' response and clarification have addressed my concerns, thus I raised my rating from 4 to 5.

**Limitations:**

See weaknesses.

**Quality:**

2

**Strengths And Weaknesses:**

Strengths:
1. From Table 2&3, the proposed method can outperform previous works on both 1.5B and 7B models and exhibits better stability in training than previous works.

Weaknesses:
1. There are too many equations that make the paper hard to follow, especially there are many symbols that do not have explaination and requires the readers to guess. For example, function $g(o,q)$ is a one time symbol that can be essentially replaced by $f(x, y)$, $f^{+}$ and $f^{-}$ can be omitted. There are also many abbreviations that show up without explanation, such as VPG, GRPO_RW in the equations.
2. The claimed motivation that previous methods have difficulty bias seems to be a bit emperical and does not have theoretical support.
3. In Table 3, when the initial policy is strong, the improvements of DisCO is small, and the baseline methods such as GRPO and DAPO even decrease the performance of initial policy, is there any explanation?

---

> ### Author Rebuttal · Authors · 2025-07-30
>
> We thank the reviewer for providing constructive comments. Below we would like to answer the raised questions.
>
> **Q1:** On the math symbols that do not have explanation. For example, function $g(o,q)$ is a one time symbol that can be essentially replaced by $f(x,y)$, $f^+$ and $f^-$ can be omitted. There are also many abbreviations that show up without explanation, such as VPG, GRPO_RW in the equations.
>
> **A:** We apologize for the confusion!
> - $g(o,q)$ in line 134 is any function, introduced for a statement of a general fact in equation (1), which is used in the proof of Proposition 1. We will move this to the appendix in the revision.
> - $f^+$ and $f^-$ in Proposition 1 are assumed to be non-decreasing functions for decomposing $f(x, y)$. Their specific forms for GRPO are actually the form in $s_{\theta}^+(o, q)$ and $s_{\theta}^-(o, q)$ in Proposition 1, i.e.,  $f^+(x, 1)=\min(x, 1+\epsilon)$ and $f^-(x,1)=\min(x, 1-\epsilon)$.   For other methods, their specific forms can be found in Table 4. We will make it  clearer in the revision.
>
> - VPG stands for Vanilla Policy Gradient method, and GRPO_RW stands for a GRPO version with the question-level weight bias removed (line 183). We will revise the paper to make the term clearer.
>
>
>
>
> **Q2:** The claimed motivation that previous methods have difficulty bias seems to be a bit empirical and does not have theoretical support.
>
> **A:** We politely disagree. Our finding regarding the difficulty bias of GRPO is through a theoretical analysis of GRPO's objective by connecting it with a discriminative objective in Proposition 1. The empirical studies in Figure 1 are used for further supporting our claim.
>
>
> **Q3:** In Table 3, when the initial policy is strong, the improvements of DisCO is small, and the baseline methods such as GRPO and DAPO even decrease the performance of initial policy, is there any explanation?
>
>
> **A:** There is a misunderstanding in the interpretation of comparison with the initial policy in Table 3.   It is important to compare their performance against the **second entry** in Table 3: *DS-Distill-Qwen-7B (32k+/8k)*, which represents the initial policy evaluated with the same 8k response length used for GRPO, DAPO and DisCO. The **first entry**, *DS-Distill-Qwen-7B (32k+/32k)*, uses a much longer 32k response length at inference and is included as a reference to highlight DisCO’s efficiency under **smaller generation budgets** (8k during both training and inference).
>
>
> In fact, both GRPO and DAPO **do improve** upon the initial policy when evaluated under the **same response length constraint**—raising performance from 0.513 to 0.592 (GRPO) and 0.570 (DAPO), respectively. Also, the improvements of DisCO are still significant, from 0.513 to 0.627 or 0.625.
>
> Please also note that while the performance of GRPO and DAPO has already saturated, DisCO continues to exhibit an upward trend as the number of training steps increases (Figure 2c).
>
>
> **Q4:** The equation 9 seems to have very similar formulation as Direct Policy Optimization. Is there any essential connection and difference between the proposed DisCO and online DPO?
>
> **A:** Indeed DPO's objective can be regarded as a discriminative objective and can be recovered in (9) by choosing $s_{\theta}(o,q) = \beta\log\frac{\pi_{\theta}(o|q)}{\pi_{ref}(o|q)}$, $\ell(s) = \log(1+\exp(-s))$, and removing the squared hinge penalty function. Nevertheless, the objective in (10) for DisCO is different from DPO, as it has an effect on weighting more on hard negative rollouts.
>
>
>
>
> **Q5:**  Is DisCO still effective and better than previous methods on other datasets like those provided in DAPO or OREAL papers?
>
> **A:** We thank the reviewer for the constructive comment. We conducted additional experiments on DAPO-Math-17k dataset with 1.5B models, training them for 1400 steps. The results are summarized below. We can observe that DisCO methods achieve slightly better performance than those trained on DeepScaleR-Preview-Dataset, consistently outperforming GRPO and other baselines.
>
> | Model                | MRL(Train/Test) | AIME 2024 | AIME 2025 | MATH 500 | AMC 2023 | Minerva | O-Bench | Avg.  |
> | -------------------- | --------------- | --------- | --------- | -------- | -------- | ------- | ------- | ----- |
> | DS-Distill-Qwen-1.5B | 32k+   /   8k   | 0.181     | 0.215     | 0.758    | 0.515    | 0.237   | 0.353   | 0.376 |
> | GRPO                 | 8k   /    8k    | 0.342     | 0.256     | 0.842    | 0.672    | 0.267   | 0.458   | 0.473 |
> | DAPO                 | 8k   /   8k     | 0.275     | 0.229     | 0.812    | 0.653    | 0.256   | 0.441   | 0.444 |
> | TRPA                 | 8k   /   8k     | 0.346     | 0.279     | 0.836    | 0.683    | 0.281   | 0.450   | 0.479 |
> | DisCO (L-ratio)      | 8k   /   8k     | 0.413     | 0.310     | **0.874**    | **0.775**    | 0.307   | 0.495   | 0.529 |
> | DisCO (log-L)        | 8k   /   8k     | **0.460**     | **0.317**     | 0.873    | **0.775**    | **0.320**   | **0.502**   | **0.541** |

---

> > ### Author Response · Authors · 2025-08-05
> > **Hope to discuss with you!**
> >
> > Dear Reviewer Hi9A,
> >
> > Thank you for your thoughtful review and encouraging rating!
> >
> > As the author-reviewer discussion period is nearing its end, we would like to follow up to see if our rebuttal has addressed your concerns. Please let us know if any further clarification would be helpful.
> >
> > Thank you again!
> >
> > Authors

---

### Official Review · Reviewer_2CBw · 2025-07-07

**Clarity:** 3
**Significance:** 2
**Originality:** 1
**Rating:** 5
**Confidence:** 4

**Summary:**

This paper proposes several modifications to GRPO:
- Does not clip advantage
- Removes down-weighting of questions with very high or low reward
- More stable KL divergence term using a squared-hinge penalty function

They train and compare their method (Disco) with GRPO, Dr. GRPO, DAPO, TRPA. They also compare against Deepseek-R1-1.5B PReview and OpenAI-o1-Preview. Disco trained model outperforms these variants on math tasks.

**Questions:**

- I'm curious what the interplay is between the expected reward and entropy collapse in your algorithm. These concepts seem to be closely related, since entropy collapse would also lead to a bimodal expected reward distribution (e.g. you could be oversampling one particular incorrect or correct answer and get extremely low/high rewards, respectively). If you reduce entropy collapse, I imagine less datapoints would have extremely low or high rewards. Yet Disco would also precisely focus more on these outlier examples. I wonder if some effects of these two design choices are somewhat cancelling each other out?
- If you remove the clipping for GRPO + use the same squared-hinged KL  term, does the performance improve?

**Ethical Concerns:**

["NO or VERY MINOR ethics concerns only"]

**Final Justification:**

From what I understand, clipping stabilizes training but performance degrades. Doing non-clipping but replacing the KL regularization term to stabilize training achieves better performance.

I think the experiments addresses my concerns. I increase my score to 5.
I think the organization of the paper could possibly improve to communicate why changing each component of GRPO was necessary.

**Limitations:**

- Based on the insights provided, are there any particular datasets you would foresee Disco to be particularly effective or suboptimal in comparison to GRPO? e.g. perhaps you can try comparing these algorithms on  datasets with a small number of very easy and hard examples.

**Quality:**

3

**Strengths And Weaknesses:**

- [Good performance] Based on numbers reported, it looks like the algorithm works well.
- [Organization needs some work. Contributions are weakly communicated --- multiple things happening with the algorithm, might be good to decompose the paper into separate sections] There seems to be two things the paper is trying to communicate: that GRPO is downweighting examples with extreme rewards + standard clipping/KL divergence term causes instability. However, in the experiments, it's very unclear where the gains are actually coming from. It might be good to more closely decompose the effect of each design choice to the performance in the main body + provide more extensive ablation studies that demonstrate the problem tackled.
- [Concurrent work] The idea of upsampling examples with the highest and lowest rewards is also explored in concurrent work Not All Rollouts are Useful [https://arxiv.org/abs/2504.13818].
- [No hyperparameter tuning for baselines] Based on the experimental section, it looks like hyperparameters are manually tuned for their algorithm, but baseline hyperparameters are reported numbers adapted from previous papers. It's not clear whether these hyperparameters are optimal for their setup.

---

> ### Author Rebuttal · Authors · 2025-07-30
>
> We thank the reviewer for providing a comprehensive review. Below we would like to address raised concerns.
>
> **Q1:** In the experiments, it might be good to more closely decompose the effect of each design choice on the performance.
>
> **A**:  We thank the reviewer for the constructive suggestion! The proposed method incorporates three key design choices:
> (1) removing question-level weight bias,
> (2) using a non-clipping scoring function, and
> (3) applying a KL-divergence constraint.
>
> The effectiveness of (1) has been demonstrated in the comparison between GRPO and GRPO$_\text{rw}$, as shown in Fig. 1 (c, d). Additionally, we conducted new experiments on training a 1.5B model, comparing DisCO with it variant that includes question-level weight bias while keeping all other components unchanged. The results are presented in the table below.
>
> The benefit of (2) is validated in Figure 3 (second from the left), which shows that using the non-clipping scoring functions yields significant improvements over its clipping-based counterpart, when all other components are the same.
>
> To evaluate the impact of (3), we conducted another experiment on the 1.5B model, comparing DisCO with a variant that replaces the KL constraint with KL-divergence regularization. The results are also summarized in the table below.
>
> We can see that each component contributes to the improvements, where the use of non-clipping scoring function contributes the most. We will incorporate the discussion into our revision.
>
> | Method                     | [pass@1(Avg.)](mailto:pass@1(Avg.)) | Method                     | [pass@1(Avg.)](mailto:pass@1(Avg.)) |
> | ------------------------- | -----------------------------------  | ------------------------- | ----------------------------------- |
> | DisCO (log-L)             | 0.533                                | DisCO (L-ratio)           | 0.524                               |
> | DisCO --> w/ weight bias            | 0.522 (↓ 0.011)            | DisCO --> w/ weight bias             | 0.511  (↓ 0.013)          |
> | DisCO -->Clipping scoring  func. | 0.430  (↓ 0.103)              | DisCO --> Clipping scoring func.| 0.430  (↓ 0.094)                |
> | DisCO--> KL Regularization          | 0.519    (↓ 0.014)                   | DisCO -->  KL Regularization         | 0.523 (↓ 0.001)             |
>
>
> **Q2:** The idea of upsampling examples with the highest and lowest rewards is also explored in concurrent work [r1].
>
>
> **A:** Thank you for pointing out the interesting work. While [r1] introduces max variance down-sampling to train on only an informative subset, we'd like to clarify that our method **does not** apply up-sampling or down-sampling to rollouts. Instead, we proposed a discriminative objective using all generated rollouts for more effective learning than GRPO. It would be interesting to integrate [r1]'s sampling approach into our framework.
>
>
> **Q3:**  Based on the experimental section, it looks like hyperparameters are manually tuned for their algorithm, but baseline hyperparameters are reported numbers adapted from previous papers. It's not clear whether these hyperparameters are optimal for their setup.
>
> **A:**  Thank you for the thoughtful comment! We would like to provide further clarification on our hyperparameter tuning process.
>
> **First**, we tune the learning rate for **all methods** from the set $\{1\text{e}{-6}, 2\text{e}{-6}\}$ (see lines 310–311).
>
> **Second**, for our proposed DisCO methods, we tune only one additional parameter, $\tau$, over a small range of three values per variant. The other hyperparameters, $\beta$ and $\delta$, are selected based on heuristics and are not tuned (lines 318-319). Our ablation study (Figure 3, right) shows that the performance is not sensitive to the choice of $\tau$.
>
> **Third**, for the baseline methods GRPO, GRPO-ER, TRPA, we follow the hyper-parameter settings used in prior works [r2, r3, r4] because they have used exactly the same model and dataset as our experiments. Dr. GRPO does not introduce any additional hyperparameters beyond the learning rate. For DAPO, we used $\epsilon_{\text{high}} = 0.28$, as in the original paper. While we acknowledge that some studies may use a different value, we tested an alternative setting $\epsilon_{\text{high}} = 0.24$ to address this concern. The result, shown in the table below, shows that $\epsilon_{\text{high}} = 0.24$ performs even worse than $\epsilon_{\text{high}} = 0.28$.
>
> In summary, the improved performance of **DisCO** does not stem from extensive or unfair hyperparameter tuning.
>
> | Model           | AIME 2024 | AIME 2025 | MATH 500 | AMC 2023 | Minerva | O-Bench | Avg.  |
> | --------------- | --------- | --------- | -------- | -------- | ------- | ------- | ----- |
> | GRPO            | 0.277     | 0.242     | 0.838    | 0.647    | 0.276   | 0.462   | 0.457 |
> | DAPO($\epsilon_{high}$ =0.24)       | 0.292     | 0.246     | 0.838    | 0.665    | 0.288   | 0.464   | 0.465 |
> | DAPO($\epsilon_{high}$ =0.28)        | 0.310     | 0.252     | 0.848    | 0.675    | 0.296   | 0.456   | 0.473 |
> | DisCO (log-L)   | 0.404     | 0.317     | 0.876    | 0.758    | 0.333   | 0.509   | 0.533 |
>
>
> **Q4:**  On the interplay  between the expected reward and entropy collapse in your algorithm. Are there less datapoints would have extremely low or high rewards in DisCO?  Would Disco focus more on these outlier examples?
>
> **A:** We would like to clarify a misunderstanding regarding DisCO. We agree that **entropy collapse** did contribute to saturation of expected rewards for GRPO, GRPO-ER and Dr. GRPO as observed in Fig. 2. Regarding DisCO,
> - **(1)** the generation entropy remains stable while the expected reward keeps increasing (see Fig. 2);
> - **(2)** We verified the reward distribution for questions in our method averaged over the first 200 steps. As shown in the following table, it does not exhibit fewer datapoints with extremely low or high rewards. It is similar to that in GRPO (Fig. 1, second from the left).
> - **(3)** It is **not true** that DisCO focuses more on the examples with extremely low or high rewards.  Indeed, DisCO is motivated by giving the same weight to all questions regardless of their expected rewards. Our objective in (10) has an effect of weighting negative rollouts for each question, focusing more on hard negative rollouts.
>
>
> | Question Accuracy | 0%   | (0, 25%] | (25%,50%] | (50%,75%] | (75%,100%) | 100% |
> | ----------------- | ---- | --------- | --------- | --------- | ---------- | ---- |
> | Ratio             | 0.24 | 0.13      | 0.11      | 0.13      | 0.1        | 0.29 |
>
>
> **Q5:** If you remove the clipping for GRPO + use the same squared-hinged KL term, does the performance improve?
>
> **A**: We have experimented with the suggested variant of GRPO, keeping its advantage function, but replacing its clipping-based scoring function with non-clipping based L-ratio, and using the squared-hinge KL penalty instead of its original KL regularization. This indeed is the variant of the second row (right) in the first Table of this rebuttal. We summarize the results below, from which we can see that its average performance 0.511 is better than GRPO (0.457) but still worse than DisCO (0.524).
> | Model            | [pass@1(Avg.)](mailto:pass@1(Avg.)) |
> | ---------------- | ----------------------------------- |
> | GRPO             | 0.457                               |
> |  No Clipped GRPO + squared-hinged KL | 0.511                               |
> | DisCO (L-ratio)  | 0.524                               |
>
>
> **Q6:** Based on the insights provided, are there any particular datasets you would foresee Disco to be particularly effective or suboptimal in comparison to GRPO? e.g. perhaps you can try comparing these algorithms on datasets with a small number of very easy and hard examples.
>
> **A:** The DeepScaleR-Preview-Dataset used in our experiments includes a wide range of questions, spanning from easy to challenging for the base models. We believe that the improvements achieved by DisCO are fundamental, rather than relying on specific properties of the dataset. To further support this claim, we conducted additional experiments on the DAPO-Math-17K dataset [r5] using 1.5B models, training them for 1400 steps. As shown below, DisCO still consistently outperforms GRPO and other baselines.
>
> | Model                | MRL(Train/Test) | AIME 2024 | AIME 2025 | MATH 500 | AMC 2023 | Minerva | O-Bench | Avg.  |
> | -------------------- | --------------- | --------- | --------- | -------- | -------- | ------- | ------- | ----- |
> | DS-Distill-Qwen-1.5B | 32k+   /   8k   | 0.181     | 0.215     | 0.758    | 0.515    | 0.237   | 0.353   | 0.376 |
> | GRPO                 | 8k   /    8k    | 0.342     | 0.256     | 0.842    | 0.672    | 0.267   | 0.458   | 0.473 |
> | DAPO                 | 8k   /   8k     | 0.275     | 0.229     | 0.812    | 0.653    | 0.256   | 0.441   | 0.444 |
> | TRPA                 | 8k   /   8k     | 0.346     | 0.279     | 0.836    | 0.683    | 0.281   | 0.450   | 0.479 |
> | DisCO (L-ratio)      | 8k   /   8k     | 0.413     | 0.310     | **0.874**    | **0.775**    | 0.307   | 0.495   | 0.529 |
> | DisCO (log-L)        | 8k   /   8k     | **0.460**     | **0.317**     | 0.873    | **0.775**    | **0.320**   | **0.502**   | **0.541** |
>
>
> [r1] Not All Rollouts are Useful: Down-Sampling Rollouts in LLM Reinforcement Learning
>
> [r2] An empirical study on eliciting and improving r1-like reasoning models.
>
> [r3] Deepscaler: Surpassing o1-preview with a 1.5b model by scaling rl.
>
> [r4] Trust region preference approximation: A simple and stable reinforcement learning algorithm for llm reasoning.
>
> [r5] DAPO: An Open-Source LLM Reinforcement Learning System at Scale.

---

> > ### Author Response · Authors · 2025-08-05
> > **Hope to discuss!**
> >
> > Dear Reviewer 2CBw,
> >
> > Thank you for your thoughtful review and encouraging rating!
> >
> > As the author-reviewer discussion period is nearing its end, we would like to follow up to see if our rebuttal has addressed your concerns. Please let us know if any further clarification would be helpful.
> >
> > Thank you again!
> >
> > Authors

---

> > ### Comment · Reviewer_2CBw · 2025-08-07
> > **Thanks!**
> >
> > Thanks for the thorough response!
> >
> > From what I understand, clipping stabilizes training but performance degrades. Doing non-clipping but replacing the KL regularization term to stabilize training achieves better performance.
> >
> > I think the experiments addresses my concerns. I'll increase my score to 5, thanks.

---

### Author Response · Authors · 2025-08-07
**Discussion in last two days**

Dear AC and all reviewers,

We are grateful to reviewers CL8D and ewTG for acknowledging our rebuttal and recommending acceptance.

To reviewers 2CBw and Hi9A: if you have any further questions or concerns regarding our rebuttal, we would be happy to engage in additional discussion before the discussion period ends.

To AC: we kindly ask for your assistance in inviting reviewers 2CBw and Hi9A to revisit our rebuttal. We hope that their evaluations will reflect the most up-to-date information.

Thank you!

Authors

---

### Decision · Program_Chairs · 2025-09-17

**Decision:**

Accept (poster)

**Comment:**

This paper proposes DisCO (Discriminative Constrained Optimization), a reinforcement learning method for large reasoning models that alleviates difficulty bias and training instability in existing methods like GRPO.

During the discussion period, the authors addressed all major concerns through comprehensive experiments and clarifications:

1. Ablation studies and component contributions (Reviewers 2CBw, CL8D). The authors provided detailed ablation results showing each component's contribution: removing question-level weight bias, using non-clipping scoring functions (the largest contributor), and applying KL constraints. These experiments clearly decompose the effectiveness of each design choice.
2. Hyperparameter tuning fairness (Reviewer 2CBw). The authors clarified that they tuned the learning rate for all methods and one additional parameter without extensive grid search for their own method DisCO. Furthermore, baselines were applied under similar experimental settings by prior works, so it was fair for the authors to follow the training settings.
3. Theoretical clarity and notation (Reviewer Hi9A). The authors clarified mathematical notation, explained the theoretical basis for difficulty bias through Proposition 1, and corrected the interpretation of Table 3 results.

All four reviewers converged on acceptance after the rebuttal. The recommendation is to accept this paper.